 TOOLS AND RESOURCES

# The gene expression landscape of the human locus coeruleus revealed by single-nucleus and spatially-resolved transcriptomics

Lukas M Weber[1†], Heena R Divecha[2†], Matthew N Tran[2], Sang Ho Kwon[2,3], Abby Spangler[2], Kelsey D Montgomery[2], Madhavi Tippani[2], Rahul Bharadwaj[2], Joel E Kleinman[2,4], Stephanie C Page[2], Thomas M Hyde[2,4,5], Leonardo Collado-Torres[2], Kristen R Maynard[2,4], Keri Martinowich[2,3,4,6*], Stephanie C Hicks[1*]

[1]Department of Biostatistics, Johns Hopkins Bloomberg School of Public Health, Baltimore, United States; [2]Lieber Institute for Brain Development, Johns Hopkins Medical Campus, Baltimore, United States; [3]Department of Neuroscience, Johns Hopkins School of Medicine, Baltimore, United States; [4]Department of Psychiatry and Behavioral Sciences, Johns Hopkins School of Medicine, Baltimore, United States; [5]Department of Neurology, Johns Hopkins School of Medicine, Baltimore, United States; [6]The Kavli Neuroscience Discovery Institute, Johns Hopkins University, Baltimore, United States

*For correspondence:
keri.martinowich@libd.org (KM);
shicks19@jhu.edu (SCH)

†These authors contributed equally to this work

Competing interest: The authors declare that no competing interests exist.

**Abstract** Norepinephrine (NE) neurons in the locus coeruleus (LC) make long-range projections throughout the central nervous system, playing critical roles in arousal and mood, as well as various components of cognition including attention, learning, and memory. The LC-NE system is also implicated in multiple neurological and neuropsychiatric disorders. Importantly, LC-NE neurons are highly sensitive to degeneration in both Alzheimer's and Parkinson's disease. Despite the clinical importance of the brain region and the prominent role of LC-NE neurons in a variety of brain and behavioral functions, a detailed molecular characterization of the LC is lacking. Here, we used a combination of spatially-resolved transcriptomics and single-nucleus RNA-sequencing to characterize the molecular landscape of the LC region and the transcriptomic profile of LC-NE neurons in the human brain. We provide a freely accessible resource of these data in web-accessible and downloadable formats.

## eLife assessment

This is an **important** initial study of cell type and spatially resolved gene expression in and around the locus coeruleus, the primary source of the neuromodulator norepinephrine in the human brain. The data are generated with cutting-edge techniques, and the work lays the foundation for future descriptive and experimental approaches to understand the contribution of the locus coeruleus to healthy brain function and disease. The empirical support for the main conclusions is **solid**. This paper, and the associated web application, will be of great interest to neuroscientists working on arousal-based behaviors and neurological and neuropsychiatric phenotypes.

## Introduction

The LC is a small bilateral nucleus located in the dorsal pons of the brainstem, which serves as the brain's primary site for the production of the neuromodulator NE. NE-producing neurons in the LC project widely to many regions of the central nervous system to modulate a variety of highly divergent functions including attention, arousal, pain, mood, and the response to stress (*Poe et al., 2020*; *Chandler et al., 2019*; *Sara, 2009*; *Morris et al., 2020*; *Suárez-Pereira et al., 2022*; *Ross and Van Bockstaele, 2020*). The LC, translated as 'blue spot' (in recognition of its characteristic presence of neuromelanin pigment), comprises merely 3000 NE neurons in the rodent (~1500–1600 on each side of the brainstem) (*Poe et al., 2020*), and estimates in the human LC range from 19,000 to 46,000 total NE neurons (*German et al., 1988*). Despite its prominent involvement in a number of critical brain functions and its distinctive capacity to synthesize NE, the LC's small size and deep positioning within the brainstem have rendered it relatively intractable to a comprehensive cellular, molecular, and physiological characterization.

The LC plays important roles in core behavioral and physiological brain function across the lifespan and in disease. Consistent with these roles, the LC-NE system is implicated in many neurodegenerative, neuropsychiatric, and neurological disorders (*Morris et al., 2020*; *Weinshenker, 2018*). The LC is one of the earliest sites of degeneration in both Alzheimer's disease (AD) and Parkinson's disease (PD), and profound loss of LC-NE neurons is evident with disease progression (*Weinshenker, 2018*; *Mather and Harley, 2016*; *Chalermpalanupap et al., 2017*). Moreover, maintaining the neural density of LC-NE neurons prevents cognitive decline during aging (*Wilson et al., 2013*). In addition, primary neuropathologies for AD (hyperphosphorylated tau) and PD (alpha-synuclein) can be detected in the LC prior to other brain regions (*Grudzien et al., 2007*; *Andrés-Benito et al., 2017*; *Braak and Del Tredici, 2012*; *Del Tredici and Braak, 2013*). However, the molecular mechanisms rendering LC-NE neurons particularly vulnerable to neurodegeneration are not well-understood. In addition to its role in neurodegenerative disorders, the LC-NE system is implicated in a number of other complex brain disorders. Noradrenergic signaling controls many cognitive functions, including sustained attention, and its dysregulation is associated with attention-deficit hyperactivity disorder (ADHD) (*Asherson et al., 2016*; *Biederman and Spencer, 1999*). Related to these findings, the NE reuptake inhibitor atomoxetine is the first non-stimulant medication that is FDA-approved for treating ADHD (*Bouret and Sara, 2004*; *Bymaster et al., 2002*; *Newman et al., 2008*). Disruption of noradrenergic signaling is also associated with anxiety, addiction, and responses to stress and trauma, and drugs that modulate noradrenergic signaling have been used in the treatment of post-traumatic stress disorder (PTSD), major depressive disorder (MDD), anxiety, and opioid withdrawal (*Morris et al., 2020*; *Dell'Osso et al., 2011*; *Weinshenker and Schroeder, 2007*; *Berridge and Waterhouse, 2003*; *Urits et al., 2020*; *Paiva et al., 2021*). Given the wide range of functions that are modulated by the LC-NE system, an improved understanding of the gene expression landscape of the LC and the surrounding region and delineating the molecular profile of LC-NE neurons in the human brain could facilitate the ability to target these neurons for disease prevention or manipulate their function for treatment in a variety of disorders.

The recent development of single-nucleus RNA-sequencing (snRNA-seq) and spatially-resolved transcriptomics (SRT) technological platforms provides an opportunity to investigate transcriptome-wide gene expression at cellular and spatial resolution (*Kamath et al., 2022*; *Maynard et al., 2021*). SRT has recently been used to characterize transcriptome-wide gene expression within defined neuroanatomy of cortical regions in the postmortem human brain (*Maynard et al., 2021*), while snRNA-seq has been used to investigate specialized cell types in a number of postmortem human brain regions including medium spiny neurons in the nucleus accumbens and dopaminergic neurons in the midbrain (*Kamath et al., 2022*; *Tran et al., 2021*). Importantly, snRNA-seq and SRT provide complementary views: snRNA-seq identifies transcriptome-wide gene expression within individual nuclei, while SRT captures transcriptome-wide gene expression in all cellular compartments (including the nucleus, cytoplasm, and cell processes) while retaining the spatial coordinates of these measurements. While not all SRT platforms achieve single-cell resolution, depending on the technological platform and tissue cell density, spatial gene expression has been resolved at, for example, ~1–10 cells per spatial measurement location with a diameter of 55 μm in the human brain (*Maynard et al., 2021*). These platforms have been successfully used in tandem to spatially map single-nucleus gene expression in several regions of both neurotypical and pathological tissues in the human brain including the

dorsolateral prefrontal cortex (*Maynard et al., 2021*) and the dopaminergic substantia nigra (*Kamath et al., 2022*).

In this report, we characterize the gene expression signature of the LC and surrounding region at spatial resolution, and identify and characterize a population of NE neurons at single-nucleus resolution in the neurotypical adult human brain. In addition to NE neurons, we identify a population of 5-hydroxytryptamine (5-HT, serotonin) neurons, which have not previously been characterized at the molecular level in human brain samples (*Okaty et al., 2019*). We observe the expression of cholinergic marker genes within NE neurons, a finding which we confirm using multiplexed single-molecule fluorescence in situ hybridization (smFISH) with high-resolution imaging at cellular resolution. We compare our findings from the human LC and adjacent region to molecular profiles of LC and peri-LC neurons that were previously characterized in rodents using alternative technological platforms (*Mulvey et al., 2018*; *Grimm et al., 2004*; *Luskin et al., 2023*), and observe partial conservation of LC-associated genes across these species.

## Results

### Experimental design and study overview of postmortem human LC

We selected five neurotypical adult human brain donors to characterize transcriptome-wide gene expression within the LC at spatial and single-nucleus resolution using the 10x Genomics Visium SRT (*10x Genomics, 2022a*) and 10x Genomics Chromium snRNA-seq (*10x Genomics, 2022b*) platforms (see *Supplementary file 1* for donor demographic details). After all quality control (QC) steps (described below), the final SRT and snRNA-seq datasets used for analyses consisted of samples from 4 and 3 donors, respectively. In each tissue sample, the LC was first visually identified by neuroanatomical landmarks and the presence of the pigment neuromelanin on transverse slabs of the pons (*Figure 1A*). Prior to SRT and snRNA-seq assays, we ensured that the tissue blocks encompassed the LC by probing for known NE neuron marker genes (*Counts and Mufson, 2012*). Specifically, we cut 10 µm cryosections from tissue blocks from each donor and probed for the presence of a pan-neuronal marker gene (*SNAP25*) and two NE neuron-specific marker genes (*TH* and *SLC6A2*) by multiplexed single-molecule fluorescence in situ hybridization (smFISH) using RNAscope (*Maynard et al., 2020*; *Wang et al., 2012*; *Figure 1B*). Robust mRNA signal from these markers, visualized as puncta on imaged tissue sections, was used as a quality control measure in all tissue blocks prior to proceeding with inclusion in the study and performing SRT and snRNA-seq assays.

For tissue blocks included in the study, we cut additional 10 µm tissue sections, which were used for gene expression profiling at spatial resolution using the 10x Genomics Visium SRT platform (*10x Genomics, 2022a*; *Figure 1C*). Fresh-frozen tissue sections were placed onto each of four capture areas per Visium slide, where each capture area contains approximately 5000 expression spots (spatial measurement locations with diameter 55 µm and 100 µm center-to-center, where transcripts are captured) laid out in a honeycomb pattern. Spatial barcodes unique to each spot are incorporated during reverse transcription, thus allowing the spatial coordinates of the gene expression measurements to be identified (*10x Genomics, 2022a*). Visium slides were stained with hematoxylin and eosin (H&E), followed by high-resolution acquisition of histology images prior to on-slide cDNA synthesis, completion of the Visium assay, and sequencing. For our study, 10 µm tissue sections from the LC-containing tissue blocks were collected from the five brain donors, with assays completed on 2–4 tissue sections per donor. Given the small size of the LC compared to the area of the array, tissue blocks were scored to fit 2–3 tissue sections from the same donor onto a single capture area to maximize the use of the Visium slides, resulting in a total of n=9 Visium capture areas (hereafter referred to as samples).

For 3 of the 5 donors, we cut additional 100 µm sections from the same tissue blocks to profile transcriptome-wide gene expression at single-nucleus resolution with the 10x Genomics Chromium single cell 3' gene expression platform (*10x Genomics, 2022b*; *Figure 1C*). Prior to collecting tissue sections, the tissue blocks were scored to enrich for NE neuron-containing regions. Neuronal enrichment was employed with fluorescence-activated nuclear sorting (FANS) prior to library preparation to enhance the capture of neuronal population diversity, and snRNA-seq assays were subsequently completed. *Supplementary file 1* provides a summary of SRT and snRNA-seq sample information and demographic characteristics of the donors.

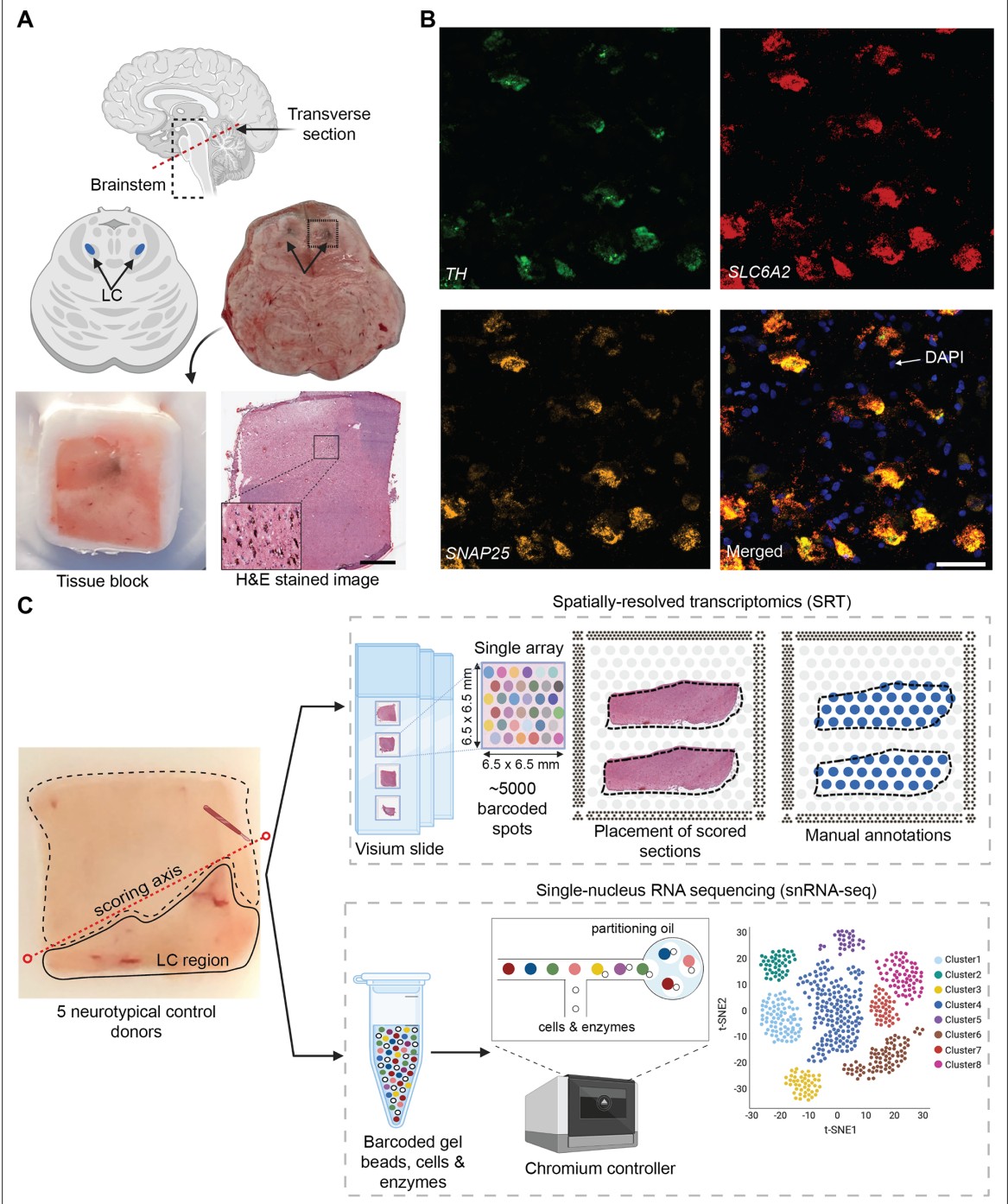

**Figure 1.** Experimental design to measure the landscape of gene expression in the postmortem human locus coeruleus (LC) using spatially-resolved transcriptomics (SRT) and single-nucleus RNA-sequencing (snRNA-seq). (**A**) Brainstem dissections at the level of the LC were conducted to collect tissue blocks from five neurotypical adult human brain donors. (**B**) Inclusion of the LC within the tissue sample block was validated using RNAscope (*Maynard et al., 2020*; *Wang et al., 2012*) for a pan-neuronal marker gene (*SNAP25*) and two norepinephrine (NE) neuron-specific marker genes (*TH* and *SLC6A2*). High-resolution hematoxylin and eosin (H&E) stained histology images were acquired prior to SRT and snRNA-seq assays (scale bars: 2 mm in H&E stained image; 20 μm in RNAscope images). (**C**) Prior to collecting tissue sections for SRT and snRNA-seq assays, tissue blocks were scored to enrich for the NE neuron-containing regions. For each sample, the LC region was manually annotated by visually identifying NE neurons in the H&E stained tissue sections. 100 μm tissue sections from three of the same donors were used for snRNA-seq assays, which included FANS-based neuronal enrichment prior to library preparation to enrich for neuronal populations. After all quality control (QC) steps, the final SRT and snRNA-seq datasets used for analyses consisted of samples from 4 and 3 donors, respectively.

## Spatial gene expression in the human LC

After applying the 10x Genomics Visium SRT platform (*10x Genomics, 2022a*), we executed several analyses to characterize transcriptome-wide gene expression at spatial resolution within the human LC. First, we manually annotated spots within regions identified as containing LC-NE neurons, based on pigmentation, cell size, and morphology from the H&E stained histology images (*Figure 2A* and *Figure 2—figure supplement 1*). Analysis of cell segmentations of the H&E images (*Figure 2—figure supplement 2A*) revealed that the median number of cells per spot within the LC regions ranged from 2 to 5 per sample (*Figure 2—figure supplement 2B*). Next, we performed additional sample-level QC on the initial n=9 Visium capture areas (hereafter referred to as samples) by visualizing the expression of two NE neuron-specific marker genes (*TH* and *SLC6A2*) (*Figure 2B*), which identified one sample (Br5459_LC_round2) without clear expression of these markers (*Figure 2—figure supplement 3A–B*). This sample was excluded from subsequent analyses, leaving n=8 samples from 4 out of the 5 donors. For the n=8 Visium samples, the annotated regions were highly enriched in the expression of the NE neuron marker genes (*TH* and *SLC6A2*) (*Figure 2C* and *Figure 2—figure supplement 3C*), confirming that these samples captured dense regions of LC-NE neurons within the annotated regions. We performed spot-level QC to remove low-quality spots based on QC metrics previously applied to SRT data (*Maynard et al., 2021*; *Amezquita et al., 2020*; *Lun et al., 2016*) (Methods). Due to the large differences in read depth between samples (*Figure 2—figure supplement 4A*, *Supplementary file 1*, Methods), we performed spot-level QC independently within each sample. After filtering low-expressed genes (Methods), this resulted in a total of 12,827 genes and 20,380 spots across the n=8 samples used for downstream analyses (*Figure 2—figure supplement 4B*).

To investigate whether the LC regions could be annotated in a data-driven manner, we applied a spatially-aware unsupervised clustering algorithm (BayesSpace, *Zhao et al., 2021*) after applying a batch integration tool (Harmony, *Korsunsky et al., 2019*) to remove sample-specific technical variation in the molecular measurements (*Figure 2—figure supplement 5*). The spatially-aware clustering using $k$=5 clusters (across eight samples) identified one cluster that overlapped with the manually annotated LC regions in several samples. However, the proportion of overlapping spots between the manually annotated LC region and this data-driven cluster (cluster 4, colored red in *Figure 2—figure supplement 6A*) was relatively low and varied across samples. We quantitatively evaluated the clustering performance by calculating the precision, recall, F1 score, and adjusted Rand index (ARI) for this cluster in each sample (see Methods for definitions). We found that while precision was >0.8 in 3 out of 8 samples, recall was <0.4 in all samples, the F1 score was <0.6 in all samples, and the ARI was <0.5 in all samples (*Figure 2—figure supplement 6B*). Therefore, we judged that the data-driven spatial domains identified from BayesSpace were not sufficiently reliable to use for the downstream analyses, and instead proceeded with the histology-driven manual annotations for all further analyses. In addition, we note that using the manual annotations avoids potential issues due to inflated false discoveries resulting from circularity when performing differential gene expression testing between sets of cells or spots defined by unsupervised clustering, when the same genes are used for both clustering and differential testing (*Gao et al., 2022*). Next, in addition to the manually annotated LC regions, we also manually annotated a set of individual spots that overlapped with NE neuron cell bodies identified within the LC regions, based on pigmentation, cell size, and morphology from the H&E histology images (*Figure 2—figure supplement 7A*). However, we observed relatively low overlap between spots with expression of NE neuron marker genes and this second set of annotated individual spots. For example, out of 706 annotated spots, only 331 spots had ≥2 observed UMI counts of *TH* (*Figure 2—figure supplement 7B*). We hypothesize that this may be due to technical factors including sampling variability in the gene expression measurements, partial overlap between spots and cell bodies, potential diffusion of mRNA molecules between spots, as well as biological variability in the expression of these marker genes. Therefore, we instead used the LC region-level manual annotations for all further analyses.

Next, to identify expressed genes associated with the LC regions, we performed DE testing between the manually annotated LC and non-LC regions by pseudobulking spots, defined as aggregating UMI counts from the combined spots within the annotated LC and non-LC regions in each sample (*Maynard et al., 2021*). This analysis identified 32 highly significant genes at a false discovery rate (FDR) threshold of $10^{-3}$ and expression FC threshold of 3 (*Figure 2D* and *Figure 2—figure supplement 8A*). This includes known NE neuron marker genes including *DBH* (the top-ranked gene

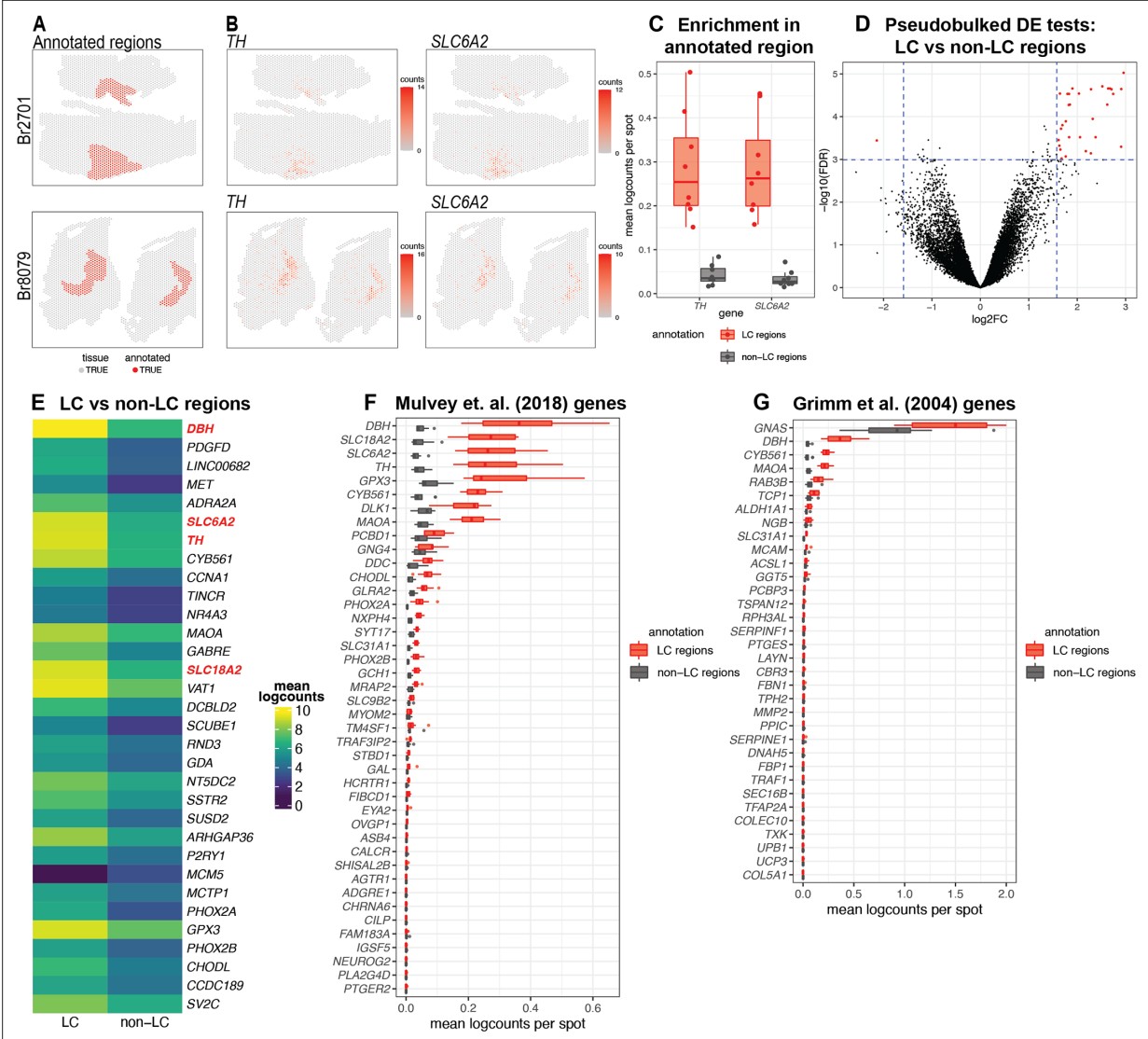

**Figure 2.** Spatial gene expression in the human locus coeruleus (LC) using spatially-resolved transcriptomics (SRT). (**A**) Spots within manually annotated LC regions containing norepinephrine (NE) neurons (red) and non-LC regions (gray), which were identified based on pigmentation, cell size, and morphology from the H&E stained histology images, from donors Br2701 (top row) and Br8079 (bottom row). (**B**) Expression of two NE neuron-specific marker genes (*TH* and *SLC6A2*). Color scale indicates unique molecular identifier (UMI) counts per spot. Additional samples corresponding to **A** and **B** are shown in **Figure 2—figure supplements 1 and 3A, B**. (**C**) Boxplots illustrating the enrichment in expression of two NE neuron-specific marker genes (*TH* and *SLC6A2*) in manually annotated LC regions compared to non-LC regions in the n=8 Visium samples. Values show mean log-transformed normalized counts (logcounts) per spot within the regions per sample. Additional details are shown in **Figure 2—figure supplement 3C**. (**D**) Volcano plot resulting from differential expression (DE) testing between the pseudobulked manually annotated LC and non-LC regions, which identified 32 highly significant genes (red) at a false discovery rate (FDR) significance threshold of $10^{-3}$ and expression fold-change (FC) threshold of 3 (dashed blue lines). Horizontal axis is shown on $log_2$ scale and vertical axis on $log_{10}$ scale. Additional details and results for 437 statistically significant genes identified at an FDR threshold of 0.05 and an FC threshold of 2 are shown in **Figure 2—figure supplement 8** and **Supplementary file 2A**. (**E**) Average expression in manually annotated LC and non-LC regions for the 32 genes from **D**. Color scale shows logcounts in the pseudobulked LC and non-LC regions averaged across n=8 Visium samples. Genes are ordered in descending order by FDR (**Supplementary file 2A**). (**F–G**) Cross-species comparison showing expression of human ortholog genes for LC-associated genes identified in the rodent LC (**Mulvey et al., 2018**; **Grimm et al., 2004**) using alternative experimental technologies. Boxplots show mean logcounts per spot in the manually annotated LC and non-LC regions per sample in the human data.

The online version of this article includes the following figure supplement(s) for figure 2:

**Figure supplement 1.** Spot-plot visualizations of manually annotated Visium spots within regions identified as containing LC-NE neurons in SRT data.

**Figure supplement 2.** H&E stained histology images and number of cells per spot for SRT data.

*Figure 2 continued on next page*

*Figure 2 continued*

**Figure supplement 3.** Spatial expression of two NE neuron-specific marker genes in Visium samples for quality control (QC) in SRT data.

**Figure supplement 4.** Spot-level quality control (QC) data visualizations for Visium samples in SRT data.

**Figure supplement 5.** Dimensionality reduction embeddings before and after batch integration across Visium samples in SRT data.

**Figure supplement 6.** Identifying LC and non-LC regions in a data-driven manner by spatially-aware unsupervised clustering in SRT data.

**Figure supplement 7.** Comparison of spot-level and region-level manual annotations in SRT data.

**Figure supplement 8.** Results from differential expression (DE) analysis to identify expressed genes associated with LC regions in SRT data.

**Figure supplement 9.** Results from applying nnSVG to identify spatially variable genes (SVGs) in SRT data.

by FDR within this set), *SLC6A2* (ranked 6th), *TH* (ranked 7th), and *SLC18A2* (ranked 14th). Out of the 32 genes, 31 were elevated in expression within the LC regions, while one (*MCM5*) was depleted. The set includes one long noncoding RNA (*LINC00682*), while the remaining 31 genes are protein-coding genes (*Figure 2E* and *Supplementary file 2A*). Alternatively, using standard significance thresholds of FDR <0.05 and expression FC >2, we identified a total of 437 statistically significant genes (*Figure 2—figure supplement 8B* and *Supplementary file 2A*).

As a second approach to identify genes associated with LC-NE neurons in an unsupervised manner, we applied a method to identify spatially variable genes (SVGs), nnSVG (*Weber et al., 2023c*). This method ranks genes in terms of the strength in the spatial correlation in their expression patterns across the tissue areas. We ran nnSVG within each contiguous tissue area containing an annotated LC region for the n=8 Visium samples (a total of 13 tissue areas, where each Visium sample contains 1–3 tissue areas) and combined the lists of top-ranked SVGs for the multiple tissue areas by averaging the ranks per gene. In this analysis, we found that a subset of the top-ranked SVGs (11 out of the top 50) were highly-ranked in samples from only one donor (Br8079), which we determined was due to the inclusion of a section of the choroid plexus adjacent to the LC in these samples (based on expression of choroid plexus marker genes including *CAPS* and *CRLF1*) (*Figure 2—figure supplement 9A–C*). In order to focus on LC-associated SVGs that were replicated across samples, we excluded the choroid plexus-associated genes by calculating an overall average ranking of SVGs that were each included within the top 100 SVGs in at least 10 of the 13 tissue areas, which identified a list of 32 highly-ranked, replicated LC-associated SVGs. These genes included known NE neuron marker genes (*DBH*, *TH*, *SLC6A2*, and *SLC18A2*) as well as mitochondrial genes (*Figure 2—figure supplement 9D* and *Supplementary file 2B*).

We also compared the expression of LC-associated genes previously identified in the rodent LC from two separate studies. The first study used translating ribosomal affinity purification sequencing (TRAP-seq) using an *SLC6A2* bacTRAP mouse line to identify gene expression profiles of the trans-latome of LC neurons (*Mulvey et al., 2018*). The second study used microarrays to assess gene expression patterns from laser-capture microdissections of individual cells in tissue sections of the rat LC (*Grimm et al., 2004*). We converted the lists of rodent LC-associated genes from these studies to human orthologs and calculated the average expression for each gene within the manually annotated LC and non-LC regions. A small number of genes from both studies were highly associated with the manually annotated LC regions in the human data, including *DBH*, *TH*, and *SLC6A2* from *Mulvey et al., 2018*, and *DBH* and *GNAS* from *Grimm et al., 2004*. However, the majority of the genes from both studies were expressed at low levels in the human data, which may reflect species-specific differences in the biological function of these genes as well as differences due to the experimental technologies employed (*Figure 2F–G*).

## Single-nucleus gene expression of NE neurons in the human LC

To add cellular resolution to our spatial analyses, we characterized gene expression in the human LC and the surrounding region at single-nucleus resolution using the 10x Genomics Chromium single cell 3' gene expression platform (*10x Genomics, 2022b*) in 3 of the same neurotypical adult donors from the SRT analyses. Samples were enriched for NE neurons by scoring tissue blocks for the LC region and performing FANS to enhance the capture of neurons. After raw data processing, doublet removal using scDblFinder (*Germain et al., 2021*), and standard QC and filtering, we obtained a total of 20,191 nuclei across the three samples (7957, 3015, and 9219 nuclei respectively from donors Br2701, Br6522, and Br8079) (see *Supplementary file 1* for additional details). For nucleus-level

QC processing, we used standard QC metrics including the sum of UMI counts and detected genes (*Amezquita et al., 2020*) (see Methods for additional details). We observed an unexpectedly high proportion of mitochondrial reads in nuclei with expression of NE neuron marker genes (*DBH*, *TH*, and *SLC6A2*), which represented our rare population of interest, and hence we did not remove nuclei based on the proportion of mitochondrial reads (*Figure 3—figure supplement 1*, *Figure 3—figure supplement 2* and *Figure 3—figure supplement 3*, additional details described below).

We identified NE neuron nuclei in the snRNA-seq data by applying an unsupervised clustering workflow adapted from workflows used for snRNA-seq data in the human brain (*Tran et al., 2021*), using a two-stage clustering algorithm consisting of high-resolution *k*-means and graph-based clustering that provides sensitivity to identify rare cell populations (*Amezquita et al., 2020*). The unsupervised clustering workflow identified 30 clusters, including clusters representing major neuronal and non-neuronal cell populations, which we labeled based on the expression of known marker genes (*Figure 3A–B*). This included a cluster of NE neurons consisting of 295 nuclei (168, 4, and 123 nuclei from donors Br2701, Br6522, and Br8079, respectively), which we identified based on the expression of NE neuron marker genes (*DBH*, *TH*, and *SLC6A2*). In addition to the NE neuron cluster, we identified clusters representing excitatory neurons, inhibitory neurons, astrocytes, endothelial and mural cells, macrophages and microglia, oligodendrocytes, and oligodendrocyte precursor cells (OPCs), as well as several clusters with ambiguous expression profiles including pan-neuronal marker genes (*SNAP25* and *SYT1*) without clear expression of excitatory or inhibitory neuronal markers, which may represent damaged neuronal nuclei and debris (*Figure 3A–B* and *Figure 3—figure supplement 1*). Further evaluation of QC metrics revealed that standard QC metrics (sum of UMI counts and detected genes) for the NE neuron cluster fell within the ranges of values observed for the other neuronal and non-neuronal clusters (*Figure 3—figure supplement 2A, B*), and that the ambiguous neuronal category included a clear subset of measurements with the overall highest mitochondrial proportions and lowest number of detected genes (likely damaged nuclei and/or debris), which remained separate from the NE neuron cluster (*Figure 3—figure supplement 2C, D*), thus providing additional confidence that the high mitochondrial proportions observed for the NE neuron cluster were not due to mis-classified damaged nuclei and/or debris.

To validate the unsupervised clustering, we also applied a supervised strategy to identify NE neuron nuclei by simply thresholding on expression of NE neuron marker genes (selecting nuclei with ≥1 UMI counts of both *DBH* and *TH*). As described above, we noted a higher than expected proportion of mitochondrial reads in nuclei with expression of *DBH* and *TH*, and did not filter on this parameter during QC processing, in order to retain these nuclei (*Figure 3—figure supplement 3A, B*). This supervised approach identified 332 NE neuron nuclei (173, 4, and 155 nuclei from donors Br2701, Br6522, and Br8079, respectively), including 188 out of the 295 NE neuron nuclei identified by unsupervised clustering (*Figure 3—figure supplement 3C*). We hypothesized that the differences in nuclei that did not agree between the two approaches were due to sampling variability in the snRNA-seq measurements for these two marker genes. To confirm this, we used an alternative method (smFISH RNAscope, *Wang et al., 2012*) to assess co-localization of three NE neuron marker genes (*DBH*, *TH*, and *SLC6A2*) within individual cells on additional tissue sections from one additional independent donor (Br8689). Visualization of high-magnification confocal images demonstrated clear co-localization of these three marker genes within individual cells (*Figure 3—figure supplement 4*). Since the unsupervised clustering is based on the expression of a large number of genes and is, therefore, less sensitive to sampling variability for individual genes, we used the unsupervised clustering results for all further downstream analyses.

We performed DE testing between the neuronal clusters and identified 327 statistically significant genes with elevated expression in the NE neuron cluster, compared to all other neuronal clusters (excluding ambiguous) captured in this region, at an FDR threshold of 0.05 and FC threshold of 2. These genes include known NE neuron marker genes (*DBH*, *TH*, *SLC6A2*, and *SLC18A2*) as well as the 13 protein-coding mitochondrial genes, which are highly expressed in large, metabolically active NE neurons (*Figure 3C*, *Figure 3—figure supplement 5A*, and *Supplementary file 2C*). Compared to the LC-associated genes identified in the SRT samples, differences are expected since the snRNA-seq data contains measurements from nuclei at single-nucleus resolution, while the SRT samples contain reads from nuclei, cytoplasm, and cell processes from multiple cell populations within the annotated LC regions.

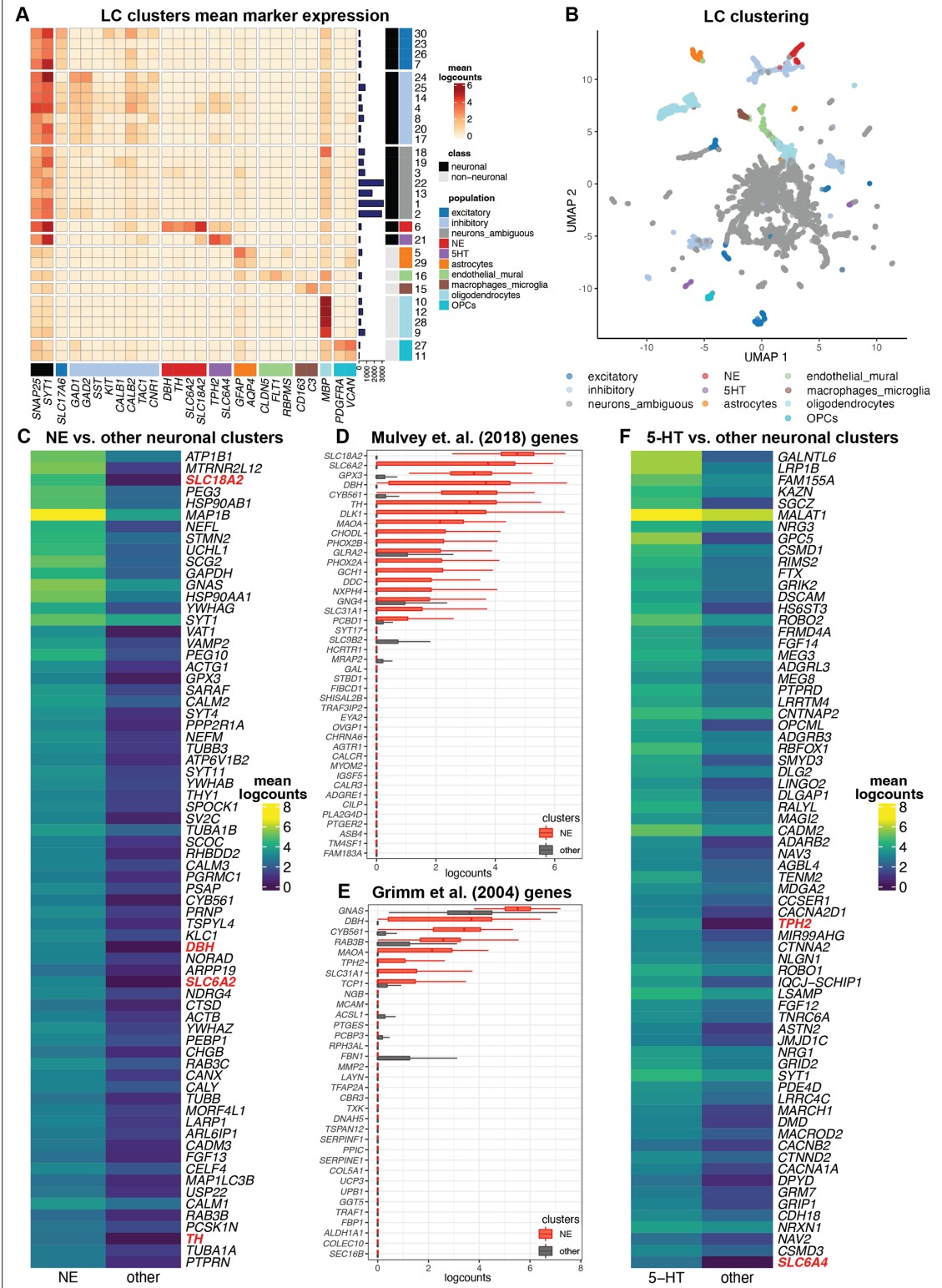

**Figure 3.** Single-nucleus gene expression in the human locus coeruleus (LC) using single-nucleus RNA-sequencing (snRNA-seq). We applied an unsupervised clustering workflow to identify cell populations in the snRNA-seq data. (**A**) Unsupervised clustering identified 30 clusters representing populations including norepinephrine (NE) neurons (red), 5-HT neurons (purple), and other major neuronal and non-neuronal cell populations (additional colors). Marker genes (columns) were used to identify clusters (rows). Cluster IDs are shown in labels on the right, and the numbers of nuclei per cluster are shown in horizontal bars on the right. Percentages of nuclei per cluster are also shown in **Figure 3—figure supplement 1D**. Heatmap

*Figure 3 continued on next page*

*Figure 3 continued*

values represent mean logcounts per cluster. (**B**) UMAP representation of nuclei, with colors matching cell populations from heatmap. (**C**) Differential expression (DE) testing between neuronal clusters identified a total of 327 statistically significant genes with elevated expression in the NE neuron cluster, at a false discovery rate (FDR) threshold of 0.05 and fold-change (FC) threshold of 2. Heatmap displays the top 70 genes, ranked in descending order by FDR, excluding mitochondrial genes, with NE neuron marker genes described in text highlighted in red. The full list of 327 genes including mitochondrial genes is provided in ***Supplementary file 2C***. Heatmap values represent mean logcounts in the NE neuron cluster and mean logcounts per cluster averaged across all other neuronal clusters (excluding ambiguous). (**D–E**) Cross-species comparison showing expression of human ortholog genes for LC-associated genes identified in the rodent LC (***Mulvey et al., 2018***; ***Grimm et al., 2004***) using alternative experimental technologies. Boxplots show logcounts per nucleus in the NE neuron cluster and all other neuronal clusters. Boxplot whiskers extend to 1.5 times the interquartile range, and outliers are not shown. (**F**) DE testing between neuronal clusters identified a total of 361 statistically significant genes with elevated expression in the 5-HT neuron cluster, at an FDR threshold of 0.05 and FC threshold of 2. Heatmap displays the top 70 genes, ranked in descending order by FDR, with 5-HT neuron marker genes described in text highlighted in red. The full list of 361 genes is provided in ***Supplementary file 2F***.

The online version of this article includes the following figure supplement(s) for figure 3:

**Figure supplement 1.** Distribution of nucleus-level quality control (QC) metrics across unsupervised clusters in snRNA-seq data.

**Figure supplement 2.** Additional quality control (QC) evaluations for NE neuron cluster in snRNA-seq data.

**Figure supplement 3.** Supervised identification of NE neuron nuclei by thresholding on expression of NE neuron marker genes in snRNA-seq data.

**Figure supplement 4.** Expression of NE neuron marker genes in individual cells using RNAscope and high-magnification confocal imaging.

**Figure supplement 5.** DE testing results between neuronal clusters in the LC and surrounding region in snRNA-seq data.

**Figure supplement 6.** DE testing results between NE neuron cluster and all other clusters in snRNA-seq data.

**Figure supplement 7.** Overlap and comparison between DE genes identified in SRT and snRNA-seq data.

**Figure supplement 8.** Unsupervised clustering results showing additional inhibitory neuronal, miscellaneous, dopaminergic, and cholinergic marker genes in snRNA-seq data.

**Figure supplement 9.** Spatial expression and enrichment analysis of 5-HT neuron marker genes in Visium SRT samples.

**Figure supplement 10.** Spatial expression of additional marker genes for 5-HT neurons in Visium SRT samples.

**Figure supplement 11.** Expression of NE neuron and 5-HT neuron marker genes using RNAscope.

**Figure supplement 12.** Expression of 5-HT neuron marker genes using RNAscope.

**Figure supplement 13.** Inhibitory neuronal subpopulations identified by secondary unsupervised clustering on inhibitory neurons in snRNA-seq data.

**Figure supplement 14.** Spot-level deconvolution to map the spatial coordinates of snRNA-seq populations within the Visium SRT samples.

**Figure supplement 15.** Spatial expression of dopamine (DA) neuron marker genes in Visium SRT samples.

**Figure supplement 16.** Expression of NE neuron marker genes, 5-HT neuron marker gene, and DA neuron marker gene in individual cells using RNAscope and high-magnification confocal imaging.

**Figure supplement 17.** High-resolution images demonstrating co-expression of cholinergic marker gene within NE neurons.

**Figure supplement 18.** Spatial expression of cholinergic marker genes in Visium SRT samples.

**Figure supplement 19.** Overview of interactive web-accessible data resources.

**Figure supplement 20.** Instructions to display individual genes in Visium SRT data app.

**Figure supplement 21.** Instructions to display individual genes in snRNA-seq data app.

To compare with the spatial data, we also performed DE testing between the NE neuron cluster and all other clusters (neuronal and non-neuronal, excluding ambiguous) in the snRNA-seq data, which identified 427 statistically significant genes with elevated expression in the NE neuron cluster at an FDR threshold of 0.05 and FC threshold of 2, including all of the 327 genes above (***Figure 3— figure supplement 6*** and ***Supplementary file 2D***). We then compared these genes with the set of 437 statistically significant DE genes from the pseudobulked analyses (LC vs. non-LC regions) in the SRT data, which generated a list of 51 genes that were identified as statistically significant DE genes in both the SRT and snRNA-seq datasets. These 51 genes included known NE neuron marker genes (*DBH*, *TH*, *SLC6A2*, and *SLC18A2*) with high FC in both datasets (***Figure 3—figure supplement 7*** and ***Supplementary file 2E***).

To compare with previous results in rodents, we evaluated the expression of the rodent LC marker genes from ***Mulvey et al., 2018*** and ***Grimm et al., 2004*** in the NE neuron cluster compared to all other neuronal clusters in the human snRNA-seq data (***Figure 3D–E***). Consistent with the SRT samples, we observed that several genes were conserved across species. However, compared to the SRT samples, we observed relatively higher expression of the conserved genes within the NE neuron

cluster, which is expected since the NE neuron cluster contains reads from individual nuclei from this population only, while the SRT data may contain reads from nuclei, cytoplasm, and cell processes from up to multiple cell populations per spot.

We note that a recent publication using snRNA-seq in mice found that LC-NE neurons were highly enriched for *Calca*, *Cartpt*, *Gal*, and *Calcr* in addition to canonical NE neuron marker genes (***Luskin et al., 2023***). In the human data, we noted significant enrichment of *GAL* and *CARTPT* in DE testing between the manually annotated LC and non-LC regions in the SRT samples (***Supplementary file 2A***). While visualization of the snRNA-seq clustering suggests that *CARTPT* is expressed in the NE neuron cluster in the snRNA-seq data (***Figure 3—figure supplement 8***), it was not identified as statistically significant in the DE testing between the NE cluster compared to all other neuronal clusters (***Supplementary file 2C***). For *CALCA* and *CALCR*, we observed no enrichment in the annotated LC regions in the SRT samples, nor in the NE neuron cluster in the snRNA-seq data (***Supplementary file 2A and 2C*** and ***Figure 3—figure supplement 8***).

## Identification of 5-HT neurons and diversity of inhibitory neuron subpopulations in single-nucleus data

In addition to NE neurons, we identified a cluster of potential 5-hydroxytryptamine (5-HT, serotonin) neurons in the unsupervised clustering analyses of the snRNA-seq data (***Figure 3A–B***) based on the expression of 5-HT neuron marker genes (*TPH2* and *SLC6A4*) (***Ren et al., 2019***). This cluster consisted of 186 nuclei (145, 28, and 13 nuclei from donors Br2701, Br6522, and Br8079, respectively). DE testing between the neuronal clusters identified 361 statistically significant genes with elevated expression in the 5-HT neuron cluster, compared to all other neuronal clusters captured in this region, at an FDR threshold of 0.05 and FC threshold of 2. These genes included the 5-HT neuron marker genes *TPH2* and *SLC6A4* (***Figure 3F***, ***Figure 3—figure supplement 5B***, and ***Supplementary file 2F***). To investigate the spatial distribution of this population, we visualized the spatial expression of the 5-HT neuron marker genes *TPH2* and *SLC6A4* in the n=9 initial Visium samples, which showed that this population was distributed across both the LC and non-LC regions (***Figure 3—figure supplement 9A, B***). Similarly, we did not observe significant spatial enrichment of *TPH2* and *SLC6A4* expression within the manually annotated LC regions in the n=8 Visium samples that passed QC (***Figure 3—figure supplement 9C***). Further analysis confirmed expression of additional marker genes for 5-HT neurons (*SLC18A2*, *FEV*) within the same regions as *TPH2* and *SLC6A4* in the Visium samples and within the 5-HT neuron cluster in the snRNA-seq data, but we did not observe consistent expression signals for *SLC6A18*, 5-HT autoreceptors (*HTR1A*, *HTR1B*), or previously identified marker genes for 5-HT neuron subpopulations within the dorsal and median raphe nuclei in mice (***Ren et al., 2019***) -- with the possible exception of the neuropeptide receptor gene *HCRTR2* (***Figure 3—figure supplements 8 and 10***). To further confirm this finding, we applied RNAscope (***Wang et al., 2012***) to visualize expression of an NE neuron marker gene (*TH*) and 5-HT neuron marker genes (*TPH2* and *SLC6A4*) within additional tissue sections from donor Br6522, which demonstrated that the NE and 5-HT marker genes were expressed within distinct cells and these neuronal populations were not localized within the same regions (***Figure 3—figure supplement 11***), and the 5-HT neuron marker genes (*TPH2* and *SLC6A4*) were co-expressed within individual cells (***Figure 3—figure supplement 12***).

We also investigated the diversity of inhibitory neuronal subpopulations within the snRNA-seq data from the human LC and the surrounding region by applying a secondary round of unsupervised clustering to the inhibitory neuron nuclei identified in the first round of clustering. This identified 14 clusters representing inhibitory neuronal subpopulations, which varied in their expression of several inhibitory neuronal marker genes including *CALB1*, *CALB2*, *TAC1*, *CNR1*, and *VIP* (additional marker genes shown in ***Figure 3—figure supplements 8 and 13A***). In addition, similar to results from a recent publication using snRNA-seq in mice (***Luskin et al., 2023***), we found that expression of neuropeptides *PNOC*, *TAC1*, and *PENK* varied across the individual inhibitory neuronal populations (***Figure 3—figure supplement 8***). We also investigated a set of GABAergic neuron marker genes from this study (***Luskin et al., 2023***) within our secondary clustering of inhibitory neurons, and observed some cluster-specific expression of several genes, including *CCK*, *PCSK1*, *PCSK2*, *PCSK1N*, *PENK*, *PNOC*, *SST*, and *TAC1* (***Figure 3—figure supplement 13B***).

In order to integrate the single-nucleus and spatial data, we also applied a spot-level deconvolution algorithm, cell2location (***Kleshchevnikov et al., 2022***), to map the spatial coordinates of NE and

5-HT neurons within the Visium samples. This algorithm integrates the snRNA-seq and SRT data by estimating the cell abundance of the snRNA-seq populations, which are used as reference populations at cellular resolution, at each spatial location (spot) in the Visium samples. We found that this approach mapped cells from the NE neuron population to the manually annotated LC regions, and cells from the 5-HT neuron population to the regions where this population was previously identified based on the expression of marker genes, although the overall mapping performance was relatively poor (*Figure 3—figure supplement 14*). In particular, we observed that cells from these neuronal populations were also mapped outside the expected regions in the Visium samples, and the estimated absolute cell abundances per spot were higher than expected. We note that these are relatively rare populations, with relatively subtle expression differences compared to other neuronal populations, and NE neurons are characterized by large size and high transcriptional activity, which may have affected the performance of the algorithm. Due to the relatively low performance, we did not rely on these results for further downstream analyses.

## Investigation of dopamine (DA) marker genes

TH is the rate-limiting enzyme in the synthesis of DA, which is used as a precursor in the synthesis of NE. Hence, expression of the *TH* gene is also used as a marker for neurons that synthesize DA, and release of DA from LC projection neurons has been observed in the rodent (*Kempadoo et al., 2016*; *Devoto and Flore, 2006*; *Devoto et al., 2005*). To evaluate whether *TH* expression in these human LC samples labels cells that produce DA or co-express DA and NE, we also investigated the expression of other genes involved in the synthesis and transport of DA. We observed minimal expression of DA neuron marker genes including *SLC6A3* (encoding the dopamine transporter), *ALDH1A1*, and *SLC26A7* within the Visium samples (*Figure 3—figure supplement 15*) and within the NE neuron cluster in the snRNA-seq data (*Figure 3—figure supplement 8*). To further validate this finding, we generated additional high-resolution RNAscope smFISH (*Wang et al., 2012*) images at 40 x magnification visualizing the expression of *DBH* and *TH* (NE neuron marker genes), *TPH2* (5-HT neuron marker gene), and *SLC6A3* (DA neuron marker gene) within individual cells in regions within the LC in samples from three independent donors (Br8689, Br5529, and Br5426). Expression of *SLC6A3* within individual NE neurons (identified by co-expression of *DBH* and *TH*) (*Figure 3—figure supplement 16*) was not apparent in these RNAscope images. Together with the RNAscope images showing co-expression of *DBH*, *TH*, and *SLC6A2* within individual NE neurons (*Figure 3—figure supplement 4*), these results strengthen the conclusion that the observed *TH*+ cells in the LC are NE-producing neurons.

## Co-expression of cholinergic marker genes within NE neurons

We observed the expression of cholinergic marker genes, including *SLC5A7*, which encodes the high-affinity choline transporter, and *ACHE*, within the NE neuron cluster in the snRNA-seq data (*Figure 3—figure supplement 8*). Because this result was unexpected, we experimentally confirmed co-expression of *SLC5A7* transcripts with transcripts for NE neuron marker genes in individual cells using RNAscope (*Wang et al., 2012*) on independent tissue sections from donors Br6522 and Br8079. We used RNAscope probes for *SLC5A7* and *TH* (NE neuron marker), and imaged stained sections at 63 x magnification to generate high-resolution images, which allowed us to definitively localize the expression of individual transcripts within cell bodies. This confirmed co-expression of *SLC5A7* and *TH*

**Table 1.** Summary of data resources providing access to datasets described in this manuscript.
All datasets described in this manuscript are freely accessible in the form of interactive web apps and downloadable R/Bioconductor objects.

| Resource | Data | Format | Link |
|---|---|---|---|
| Shiny (*Chang et al., 2019*) web app | Visium SRT | Interactive web app | https://libd.shinyapps.io/locus-c_Visium/ |
| iSEE (*Rue-Albrecht et al., 2018*) web app | snRNA-seq | Interactive web app | https://libd.shinyapps.io/locus-c_snRNA-seq/ |
| R/Bioconductor ExperimentHub data package | Visium SRT and snRNA-seq | Downloadable R/Bioconductor objects in SpatialExperiment (*Righelli et al., 2022*) and SingleCellExperiment (*Amezquita et al., 2020*) formats | https://bioconductor.org/packages/WeberDivechaLCdata |

in individual cells in a tissue section from donor Br8079 (*Figure 3—figure supplement 17*), validating that these transcripts are expressed within the same cells. To further investigate the spatial distribution of the cholinergic marker genes, we visualized the expression of *SLC5A7* and *ACHE* in the Visium samples, which showed that these genes were expressed both within and outside the annotated LC regions (*Figure 3—figure supplement 18*).

## Interactive and accessible data resources

We provide freely accessible data resources containing all datasets described in this manuscript, in the form of both interactive web-accessible and downloadable resources (*Table 1*). The interactive resources can be explored in a web browser via a Shiny (*Chang et al., 2019*) app for the Visium SRT data and an iSEE (*Rue-Albrecht et al., 2018*) app for the snRNA-seq data (*Figure 3—figure supplements 19–21*). The data resources are also available from an R/Bioconductor ExperimentHub data package as downloadable objects in the SpatialExperiment (*Righelli et al., 2022*) and SingleCellExperiment (*Amezquita et al., 2020*) formats, which can be loaded in an R session and contain metadata including manual annotation labels and cluster labels.

## Discussion

Due to the small size and inaccessibility of the LC within the brainstem, this region has been relatively understudied in the human brain, despite its involvement in numerous functions and disease mechanisms. Our dataset provides the first transcriptome-wide characterization of the gene expression landscape of the human LC using spatially-resolved transcriptomics (SRT) and single-nucleus RNA-sequencing (snRNA-seq). Analysis of these data identified a population of norepinephrine (NE) neurons as well as a population of 5-hydroxytryptamine (5-HT, serotonin) neurons, and spatially localized them within the LC and surrounding region. We evaluated the expression of previously known marker genes for these populations and identified novel sets of significant differentially expressed (DE) genes, and assessed how their expression varies in space across the neuroanatomy of the region. We evaluated the expression of dopamine (DA) marker genes to confirm that the observed *TH* expression likely maps to NE-synthesizing, as opposed to DA-synthesizing neurons. We compared our findings from the human LC to molecular profiles of LC and peri-LC neurons in rodents, and confirmed partial conservation of LC-associated genes across these species. However, we note caution in interpreting these results given differences in technological platforms and experimental approaches across studies. Finally, we validated key results with smFISH RNAscope by assessing co-localization of genes of interest with established marker genes on independent tissue sections.

Identifying genes whose expression is enriched in NE neurons is important because the LC-NE system is implicated in multiple neuropsychiatric and neurological disorders (*Morris et al., 2020*; *Weinshenker, 2018*), and prominent loss of NE cells in the LC occurs in neurodegenerative disorders (*Weinshenker, 2018*; *Mather and Harley, 2016*; *Chalermpalanupap et al., 2017*). Untargeted, transcriptome-wide analysis of the snRNA-seq and SRT data identified a number of genes that are enriched in the human LC region and in LC-NE neurons themselves. As expected, these analyses validated the enrichment of genes involved in NE synthesis and packaging (*TH*, *SLC18A2*, *DBH*) as well as NE reuptake (*SLC6A2*). We also noted expression selectively in LC-NE neurons of a number of genes whose expression is altered in animal models or in human disease in the LC (*SSTR2*, *PHOX2A*, *PHOX2B*) (*Fan et al., 2018*; *Ádori et al., 2015*). We identified LC enrichment of a number of genes that have been associated at the cellular level with apoptosis, cell loss, or pathology in the context of neurodegeneration (*RND3*, *P2RY1*) (*Pietrowski et al., 2021*; *Moore et al., 2000*; *Cueto-Ureña et al., 2022*; *Dong et al., 2021*). We also noted enrichment of *NT5DC2* in the LC, a gene which has been associated with ADHD and regulates catecholamine synthesis in vitro (*van Hulzen et al., 2017*; *Nakashima et al., 2019*). Localization of these genes to the LC in humans, and LC-NE neurons in particular, may provide important biological insights about the physiological function of these neurons and provide context about underlying mechanistic links between these genes and disease risk. Future work using the transcriptome-wide molecular expression profiles of NE neurons at single-nucleus resolution and the LC region at spatial resolution generated here could investigate associations with individual genes and gene sets from genome-wide association studies (GWAS) for these disorders as well as genes more generally associated with aging-related processes.

Our study has several limitations. The SRT data using the 10x Genomics Visium platform captured a median of 2–5 cells per measurement location within the LC regions per sample, and future studies could apply higher-resolution platforms to characterize expression at single-cell or sub-cellular spatial resolution. These estimates of cell density could also be used to estimate the expected yield of nuclei in single-nucleus data in future studies. In the single-nucleus data, we identified a relatively small number of NE neurons, which may be related to technical factors that affected the recovery of this population due to their relatively large size and fragility or contributed to damage and loss of NE neuron nuclei. For example, we opted to use fluorescence-activated nuclei sorting (FANS) to enrich for neurons based on our previous success with this approach to identify relatively rare neuronal populations in other brain regions including the nucleus accumbens and amygdala (*Tran et al., 2021*). This and other technical factors may have also contributed to the unexpectedly high proportion of mitochondrial reads that we observed in NE neurons in the snRNA-seq data. While mitochondrial reads are not expected in the nuclear compartment, recent studies reported contamination of nuclear preparations in snRNA-seq data with ambient mRNAs from abundant cellular transcripts (*Caglayan et al., 2022*). Given the relatively elevated energy demand and increased metabolic activity of NE neurons, higher than expected mitochondrial contamination in the nuclear preparation of LC tissue may be plausible. Because NE neurons were the population of highest interest to profile in this dataset, we opted not to perform QC filtering on the proportion of mitochondrial reads, in order to retain this population. Further optimizing technical procedures for cell sorting and cryosectioning to avoid cellular damage, as well as for optimal isolation of larger cell sizes, could enhance recovery of this population for future, larger-scale snRNA-seq studies in this brain region.

Since the identification of potential 5-HT neurons in the single-nucleus data was an unexpected finding, the experimental procedures were not designed to optimally recover this population, and the precise anatomical origin of the 5-HT neurons recovered in this dataset is not entirely clear. While our dissection strategy in this initial study precluded the ability to keep track of the exact orientation of the tissue sections, we have now developed a number of specialized technical and logistical strategies for keeping track of the orientation of dissections and mounting serial sections from the same tissue block onto a single array, which will make it possible to address these, and other important questions in the future. For example, interrogating spatial gradients across the rostral-caudal gradient of the LC will be important given the differential association with vulnerability to neurodegeneration across this axis (*Beardmore et al., 2021*; *Theofilas et al., 2017*). It is possible that the identified 5-HT neurons were close to the borders of the LC dissections, residing within the dorsal raphe nucleus, which is neuroanatomically adjacent to the LC. Supporting this hypothesis, RNAscope data in *Figure 3—figure supplement 11* from an independent tissue section shows that *TPH2* and *SLC6A4* expression appears to be distinct from the LC region, which is densely packed with NE cells. However, there is some evidence for the expression of serotonergic markers within the LC region in rodents (*Grimm et al., 2004*; *Iijima, 1993*; *Iijima, 1989*), and our SRT data does support this possibility in the human brain, although further characterization is needed. Comprehensively understanding the full molecular diversity of 5-HT neurons in the human brain would require dissections that systematically sample across the extent of the dorsal raphe nucleus across the rostral-caudal axis of the brainstem.

Similarly, the identification of cholinergic marker gene expression, particularly the robust expression of *SLC5A7* and *ACHE* within NE neurons, was unexpected. While previous studies have identified small populations of cholinergic interneurons within or adjacent to the LC in rodents (*Luskin et al., 2023*), analysis of our data did not classify any of the other neuronal populations as cholinergic per se. However, both the SRT and RNAscope data (*Figure 3—figure supplement 15*) support the hypothesis that expression of cholinergic markers occurs in NE cells themselves, as well as in sparse populations of cholinergic neurons adjacent to the LC region that do not express NE marker genes. We were surprised by the expression of *SLC5A7* and *ACHE* in the LC regions (Visium data) and within the LC-NE neuron cluster (snRNA-seq data), coupled with the absence of other typical cholinergic marker genes (e.g. *CHAT*, *SLC18A3*). However, previous work suggested non-cholinergic actions of *ACHE*, particularly in other catecholaminergic neuron populations (e.g. dopaminergic neurons in the substantia nigra) (*Halliday and Greenfield, 2012*; *Greenfield, 1991*), raising the possibility of a similar mechanism in the LC-NE catecholaminergic population. We note that our snRNA-seq data may be underpowered to fully identify and classify these sparse populations, and future experiments designed to specifically investigate this finding in more detail could lead to a better understanding

of cholinergic signaling within the LC of the human brain. Similarly, our snRNA-seq may be under-powered to perform gene set enrichment studies (*De Leeuw et al., 2015*) to identify whether the NE and 5-HT neurons harbor aggregated genetic risk for psychiatric disorders, but these data could be aggregated with future snRNA-seq data generated from the LC to address this question. To facilitate further exploration of these data, we provide a freely accessible data resource that includes the snRNA-seq and SRT data in both web-based and R-based formats, as well as reproducible code for the computational analysis workflows for the snRNA-seq and SRT data.

## Materials and methods

### Postmortem human brain tissue samples for RNAscope, SRT, and snRNA-seq assays

Brain donations in the Lieber Institute for Brain Development (LIBD) Human Brain Repository (HBR) were collected with audiotaped witnessed informed consent with legal next-of-kin from the Office of the Chief Medical Examiner of the State of Maryland under the Maryland Department of Health's IRB protocol #12–24, and from the Western Michigan University Homer Stryker MD School of Medicine, Department of Pathology, and the Department of Pathology, University of North Dakota School of Medicine and Health Sciences, both under WCG IRB protocol #20111080. Clinical characterization, diagnoses, and macro- and microscopic neuropathological examinations were performed on all samples using a standardized paradigm, and subjects with evidence of macro- or microscopic neuropathology were excluded. Details of tissue acquisition, handling, processing, dissection, clinical characterization, diagnoses, neuropathological examinations, RNA extraction, and quality control measures have been described previously (*Lipska et al., 2006*; *Zandi et al., 2022*). We obtained tissue blocks from five male neurotypical brain donors of European ancestry. To select tissue blocks for study inclusion, we identified the LC in transverse slabs of the pons from the fresh-frozen human brain. The LC was identified through visual inspection of the slab, based on neuroanatomical landmarks and the presence of neuromelanin pigmentation. For each donor, a tissue block was dissected from the dorsal aspect of the pons, centered around the LC, using a dental drill. The tissue block was taken at the level of the motor trigeminal nucleus and superior cerebellar peduncle. Tissue blocks were kept at –80 °C until sectioning for experiments. We cut 10 µm tissue sections for performing SRT assays using the 10x Genomics Visium SRT platform (*10x Genomics, 2022a*). High-resolution images of the H&E stained histology were acquired prior to on-slide cDNA synthesis and completing the Visium assays. Assays were performed on 2–4 tissue sections collected from each of the five donors, and the tissue blocks were scored to fit 2–3 tissue sections from the same donor onto a single Visium capture area to maximize the use of the Visium slides. This resulted in a total of n=9 Visium capture areas (hereafter referred to as samples) in the SRT dataset. For 3 of the 5 donors, we cut additional 100 µm cryosections for snRNA-seq assays using the 10x Genomics Chromium snRNA-seq platform (*10x Genomics, 2022b*). *Supplementary file 1* provides information on brain donor demographics as well as sample information for the SRT and snRNA-seq datasets.

### Multiplexed smFISH using RNAscope

For RNAscope experiments, tissue blocks were sectioned at 10 µm and single-molecule fluorescent in situ hybridization assays were performed with RNAscope technology (*Wang et al., 2012*) using the Fluorescent Multiplex Kit v.2 and 4-plex Ancillary Kit (catalog no. 323100, 323120 ACD) according to the manufacturer's instructions. Briefly, 10 µm tissue sections (2–4 sections per donor) were fixed with 10% neutral buffered formalin solution (catalog no. HT501128, Sigma-Aldrich) for 30 min at room temperature, series dehydrated in increasing concentrations of ethanol (50%, 70%, 100%, and 100%), pretreated with hydrogen peroxide for 10 min at room temperature and treated with protease IV for 30 min. For QC experiments to confirm LC inclusion in the tissue block (*Figure 1B* showing example for additional independent donor Br8689), tissue sections were incubated with three different probes (two LC-NE neuron markers and one pan-neuronal marker): *SLC6A2* (catalog no. 526801-C1, Advanced Cell Diagnostics) encoding the norepinephrine transporter, *TH* (catalog no. 441651-C2, Advanced Cell Diagnostics) encoding tyrosine hydroxylase, and *SNAP25* (catalog no. 518851-C3, Advanced Cell Diagnostics). To confirm co-expression of LC-NE marker genes within individual cells (*Figure 3—figure supplement 4*), we used *SLC6A2* (catalog no. 526801-C1, Advanced Cell Diagnostics), *TH* (catalog

no. 441651-C2, Advanced Cell Diagnostics), and *DBH* (catalog no. 545791-C3, Advanced Cell Diagnostics) encoding dopamine beta-hydroxylase. To localize serotonergic and cholinergic markers within the LC (*Figure 3—figure supplement 11*), we used *TH* (catalog no. 441651-C2, Advanced Cell Diagnostics) encoding tyrosine hydroxylase, *TPH2* (catalog no. 471451-C1, Advanced Cell Diagnostics) encoding tryptophan hydroxylase 2, *SLC6A4* (catalog no. 604741-C3, Advanced Cell Diagnostics) encoding the serotonin transporter, and *SLC5A7* (catalog no. 564671-C4, Advanced Cell Diagnostics) encoding the high-affinity choline transporter. After probe labeling, sections were stored overnight in 4 x saline-sodium citrate buffer (catalog no. 10128–690, VWR). After amplification steps (AMP1-3), probes were fluorescently labeled with Opal Dyes 520, 570, and 690 (catalog no. FP1487001KT, FP1488001KT, and FP1497001KT, Akoya Biosciences; 1:500 dilutions for all the dyes) and counterstained with DAPI (4',6-diamidino-2-phenylindole) to label cell nuclei. Lambda stacks were acquired in *z*-series using a Zeiss LSM780 confocal microscope equipped with 20 x, 0.8 numerical aperture (NA) and 63 x, 1.4 NA objectives, a GaAsP spectral detector, and 405-, 488-, 561- and 633 nm lasers. All lambda stacks were acquired with the same imaging settings and laser power intensities. After image acquisition, lambda stacks in *z*-series were linearly unmixed using Zen Black (weighted; no autoscale) using reference emission spectral profiles previously created in Zen for the dotdotdot software (git hash v.4e1350b) (*Maynard et al., 2020*), stitched, maximum intensity projected, and saved as Carl Zeiss Image (.czi) files.

## Visium SRT with H&E staining data generation and sequencing

Tissue blocks were embedded in an OCT medium and cryosectioned at 10 µm on a cryostat (Leica Biosystems). Briefly, Visium Gene Expression Slides were cooled inside the cryostat, and tissue sections were then adhered to the slides. Tissue sections were fixed with methanol and then stained with H&E according to the manufacturer's staining and imaging instructions (User guide CG000160 Rev C). Images of the H&E stained slides were acquired using a CS2 slide scanner (Leica Biosystems) equipped with a color camera and a 20 x, 0.75 NA objective and saved as a Tagged Image File (.tif). Following H&E staining and acquisition of images, slides were processed for the Visium assay according to the manufacturer's reagent instructions (Visium Gene Expression protocol User guide CG000239, Rev D) as previously described (*Maynard et al., 2021*). In brief, the workflow includes permeabilization of the tissue to allow access to mRNA, followed by reverse transcription, removal of cDNA from the slide, and library construction. Tissue permeabilization experiments were conducted on a single LC sample used in the study and an optimal permeabilization time of 18 min was identified and used for all sections across donors. Sequencing libraries were quality-controlled and then sequenced on the MiSeq, NextSeq, or NovaSeq Illumina platforms. Sequencing details and summary statistics for each sample are reported in *Supplementary file 1*.

## Number of cells per spot in Visium SRT data

We applied cell segmentation software (VistoSeg, *Tippani et al., 2023*) to the H&E stained histology images to estimate the number of cells per spot in the Visium SRT data, and calculated summary statistics for the annotated LC regions per sample, for 6 out of the 9 Visium SRT samples (*Figure 2—figure supplement 2B*). For the remaining three samples, the H&E stained histology images were of insufficient quality to reliably estimate the number of cells per spot. We applied VistoSeg to count cell bodies based on the identification of nuclei using pixel-based counting (with a minimum size threshold of 100 pixels and watershed segmentation algorithm) (*Tippani et al., 2023*).

## snRNA-seq data generation and sequencing

Following SRT data collection, three tissue blocks (Br6522, Br8079, and Br2701) were used for snRNA-seq. Prior to tissue collection for snRNA-seq assays, tissue blocks were further scored based on the RNAscope and SRT data to enrich tissue collection to the localized site of LC-NE neurons. After scoring the tissue blocks, samples were sectioned at 100 µm and collected in cryotubes. The sample collected for Br6522 contained 10 sections, weighing 51 mg, and the samples from Br8079 and Br2701 each contained 15 sections, weighing 60.9 mg and 78.9 mg, respectively. These samples were processed following a modified version of the 'Frankenstein' nuclei isolation protocol as previously described (*Tran et al., 2021*). Specifically, chilled EZ lysis buffer (MilliporeSigma) was added to the LoBind microcentrifuge tube (Eppendorf) containing cryosections, and the tissue was gently

broken up, on ice, via pipetting. This lysate was transferred to a chilled dounce, rinsing the tube with additional EZ lysis buffer. The tissue was then homogenized using part A and B pestles, respectively, for ~10 strokes, each, and the homogenate was strained through a 70 μm cell strainer. After lysis, the samples were centrifuged at 500 g at 4 °C for 5 min, the supernatant was removed, then the sample was resuspended in EZ lysis buffer. Following a 5 min incubation, samples were centrifuged again. After supernatant removal, wash/resuspension buffer (PBS, 1% BSA, and 0.2 U/uL RNasin), was gently added to the pellet for buffer interchange. After a 5 min incubation, each pellet was again washed, resuspended, and centrifuged three times.

For staining, each pellet was resuspended in a wash/resuspension buffer with 3% BSA, and stained with AF488-conjugated anti-NeuN antibody (MiliporeSigma, catalog no. MAB377X), for 30 min on ice with frequent, gentle mixing. After incubation, these samples were washed with 1 mL wash/resuspension buffer, then centrifuged, and after the supernatant was aspirated, the pellet was resuspended in wash/resuspension buffer with propidium iodide (PI), then strained through a 35 μm cell filter attached to a FACS tube. Each sample was then sorted on a Bio-Rad S3e Cell Sorter on 'Purity' mode into a 10x Genomics reverse transcription mix, without enzyme. A range of 5637–9000 nuclei were sorted for each sample, aiming for an enrichment of ~60% singlet, NeuN+ nuclei. Then the 10x Chromium reaction was completed following the Chromium Next GEM Single Cell 3' Reagent Kits v3.1 (Dual Index) revision A protocol, provided by the manufacturer (10x Genomics) to generate libraries for sequencing. The number of sequencing reads and platforms used per sample are shown in **Supplementary file 1**.

## Analysis of Visium SRT data

This section describes additional details on the computational analyses of the Visium SRT data that are not included in the main text.

Manual alignment of the H&E stained histology images to the expression spot grid was performed using the 10x Genomics Loupe Browser software (v. 5.1.0). The raw sequencing data files (FASTQ files) for the sequenced library samples were processed using the 10x Genomics Space Ranger software (v. 1.3.0) (**10x Genomics, 2022c**) using the human genome reference transcriptome version GRCh38 2020-A (July 7, 2020) provided by 10x Genomics. Sequencing summary statistics for each sample are provided in **Supplementary file 1**.

For spot-level QC filtering, we removed outlier spots that were more than three median absolute deviations (MADs) above or below the median sum of UMI counts or the median number of detected genes per spot (**Amezquita et al., 2020**). We did not use the proportion of mitochondrial reads per spot for spot-level QC filtering, since we observed a high proportion of mitochondrial reads in the NE nuclei in the snRNA-seq data (for more details, see Results and **Figure 3—figure supplement 1**), so using this QC metric would risk removing the rare population of NE nuclei of interest in the snRNA-seq data. Therefore, for consistency with the snRNA-seq analyses, we did not use the proportion of mitochondrial reads for spot-level QC filtering in the SRT data. Due to the large differences in read depth between samples (e.g. median sum of UMI counts ranged from 118 for sample Br6522_LC_2_round1 to 2,252 for sample Br6522_LC_round3; see **Figure 2—figure supplement 4A** and **Supplementary file 1** for additional details), we performed spot-level QC independently within each sample. The spot-level QC filtering identified a total of 287 low-quality spots (1.4% out of 20,667 total spots) from the n=8 Visium samples that passed sample-level QC. These spots were removed from subsequent analyses (**Figure 2—figure supplement 4B**). For gene-level QC filtering, we removed low-expressed genes with a total of less than 80 UMI counts summed across the n=8 Visium samples.

For the evaluation of the spatially-aware clustering with BayesSpace (**Zhao et al., 2021**) to identify the LC regions in a data-driven manner, precision is defined as the proportion of spots in the selected cluster that are from the true annotated LC region, recall is defined as the proportion of spots in the true annotated LC region that are in the selected cluster, F1 score is defined as the harmonic mean of precision and recall (values ranging from 0 to 1, with 1 representing perfect accuracy), and the adjusted Rand index is defined as the percentage of correct assignments, adjusted for chance (values ranging from 0 for random assignment to 1 for perfect assignment).

For the pseudobulked DE testing, we aggregated the reads within the LC and non-LC regions using the scater package (**McCarthy et al., 2017**), and then used the limma package (**Ritchie et al., 2015**) to calculate empirical Bayes moderated DE tests.

## Analysis of snRNA-seq data

This section describes additional details on the computational analyses of the snRNA-seq data that are not included in the main text.

We aligned sequencing reads using the 10x Genomics Cell Ranger software (*10x Genomics, 2022d*) (version 6.1.1, cellranger count, with option --include-introns), using the human genome reference transcriptome version GRCh38 2020-A (July 7, 2020) provided by 10x Genomics. We called nuclei (distinguishing barcodes containing nuclei from empty droplets) using Cell Ranger ('filtered' outputs), which recovered 8979, 3220, and 10,585 barcodes for donors Br2701, Br6522, and Br8079, respectively. We applied scDblFinder (*Germain et al., 2021*) using default parameters to computationally identify and remove doublets, which removed 1022, 205, and 1366 barcodes identified as doublets for donors Br2701, Br6522, and Br8079, respectively.

We performed nucleus-level QC processing by defining low-quality nuclei as nuclei with outlier values more than three MADs above or below the median sum of UMI counts or the median number of detected genes (*Amezquita et al., 2020*), which did not identify any low-quality nuclei, so all nuclei were retained. We did not use the proportion of mitochondrial reads for QC processing, since we observed a high proportion of mitochondrial reads in nuclei with expression of NE neuron markers (*DBH* and *TH*) (for more details, see Results and *Figure 3—figure supplement 3*). Therefore, QC filtering on the proportion of mitochondrial reads would risk removing the rare population of NE neuron nuclei of interest. We performed gene-level QC filtering by removing low-expressed genes with less than 30 UMI counts summed across all nuclei. After doublet removal and QC processing, we obtained a total of 7957, 3015, and 9219 nuclei from donors Br2701, Br6522, and Br8079, respectively.

For the unsupervised clustering, we used a two-stage clustering algorithm consisting of high-resolution *k*-means and graph-based clustering that provides sensitivity to identify rare cell populations (*Amezquita et al., 2020*). For the first round of clustering (results in *Figure 3A–B*, *Figure 3—figure supplement 8*), we used 2000 clusters for the *k*-means step, and 10 nearest neighbors and Walktrap clustering for the graph-based step. For the secondary clustering of inhibitory neurons (*Figure 3—figure supplement 13*), we used 1000 clusters for the *k*-means step, and 10 nearest neighbors and Walktrap clustering for the graph-based step. We did not perform any batch integration prior to clustering, since batch integration algorithms may strongly affect rare populations but these algorithms have not yet been independently evaluated on datasets with rare populations (<1% of cells). We calculated highly variable genes, log-transformed normalized counts (logcounts), and performed dimensionality reduction using the scater and scran packages (*Lun et al., 2016*; *McCarthy et al., 2017*).

For the DE testing, we performed pairwise DE testing between all neuronal clusters, using the findMarkers() function from the scran package (*Lun et al., 2016*). We tested for genes with $\log_2$-fold-changes ($\log_2$FC) significantly greater than 1 (lfc = 1, direction = 'up') to identify genes with elevated expression in any of the neuronal clusters compared to all other neuronal clusters.

For the spot-level deconvolution using cell2location (*Kleshchevnikov et al., 2022*), we used the following parameters for human brain data from the Visium platform: detection_alpha = 20, N_cells_per_location = 3.

## Code availability

Code scripts to reproduce all analyses and figures in this manuscript, including the computational analysis workflows for the snRNA-seq and SRT data, are available from GitHub at https://github.com/lmweber/locus-c (*Weber, 2023a*). We used R version 4.2 and Bioconductor version 3.15 packages for analyses in R.

## Acknowledgements

The authors would like to extend their gratitude to the families and next of kin of the donors for their generosity in supporting and expanding our knowledge of the human brain and neuropsychiatric disease. We would also like to thank the physicians and staff of the Office of the Chief Medical Examiner of the State of Maryland, the Western Michigan University Homer Stryker MD School of Medicine, Department of Pathology, and the Department of Pathology, University of North Dakota School of Medicine and Health Sciences Medical Examiners' office. We would also like to extend our appreciation to Drs. Lewellyn Bigelow and Fernando Goes, and Amy Deep-Soboslay for their provision

of the detailed diagnostic evaluation of each case used in this study, James Tooke for his assistance with coordinating dissections within the Lieber Institute for Brain Development (LIBD) Human Brain Repository, and Daniel Weinberger for suggestions and advice on the manuscript. Research reported in this publication was supported by the Lieber Institute for Brain Development, National Institutes of Health awards U01MH122849 (KM, SCH), R01DA053581 (KM, SCH), K99HG012229 (LMW), and awards CZF2019-002443 and CZF2018-183446 (SCH) from the Chan Zuckerberg Initiative DAF, an advised fund of Silicon Valley Community Foundation.

## Additional information

### Competing interests

The authors declare that no competing interests exist.

### Funding

| Funder | Grant reference number | Author |
| --- | --- | --- |
| National Institutes of Health | U01MH122849 | Keri Martinowich<br>Stephanie C Hicks |
| National Institutes of Health | R01DA053581 | Keri Martinowich<br>Stephanie C Hicks |
| National Institutes of Health | K99HG012229 | Lukas M Weber |
| Chan Zuckerberg Initiative | CZF2019-002443 | Stephanie C Hicks |
| Chan Zuckerberg Initiative | CZF2018-183446 | Stephanie C Hicks |

The funders had no role in study design, data collection and interpretation, or the decision to submit the work for publication.

### Author contributions

Lukas M Weber, Data curation, Software, Formal analysis, Funding acquisition, Visualization, Writing – original draft, Writing – review and editing; Heena R Divecha, Data curation, Software, Validation, Investigation, Visualization, Methodology, Writing – original draft, Writing – review and editing; Matthew N Tran, Data curation, Formal analysis, Investigation, Methodology, Writing – original draft, Writing – review and editing; Sang Ho Kwon, Investigation, Methodology; Abby Spangler, Kelsey D Montgomery, Investigation; Madhavi Tippani, Data curation, Visualization; Rahul Bharadwaj, Joel E Kleinman, Thomas M Hyde, Resources; Stephanie C Page, Supervision; Leonardo Collado-Torres, Software; Kristen R Maynard, Conceptualization, Supervision, Funding acquisition, Methodology, Project administration, Writing – review and editing; Keri Martinowich, Stephanie C Hicks, Conceptualization, Supervision, Funding acquisition, Writing – original draft, Project administration, Writing – review and editing

### Author ORCIDs

Lukas M Weber ![ORCID] https://orcid.org/0000-0002-3282-1730
Heena R Divecha ![ORCID] https://orcid.org/0000-0002-1959-0675
Stephanie C Page ![ORCID] https://orcid.org/0000-0002-1951-7398
Leonardo Collado-Torres ![ORCID] http://orcid.org/0000-0003-2140-308X
Kristen R Maynard ![ORCID] http://orcid.org/0000-0003-0031-8468
Keri Martinowich ![ORCID] https://orcid.org/0000-0002-5237-0789
Stephanie C Hicks ![ORCID] https://orcid.org/0000-0002-7858-0231

### Ethics

Brain donations in the Lieber Institute for Brain Development (LIBD) Human Brain Repository (HBR) were collected with audiotaped witnessed informed consent with legal next-of-kin from the Office of the Chief Medical Examiner of the State of Maryland under the Maryland Department of Health's IRB protocol #12-24, and from the Western Michigan University Homer Stryker MD School of Medicine,

Department of Pathology, and the Department of Pathology, University of North Dakota School of Medicine and Health Sciences, both under WCG IRB protocol #20111080.

Reviewer #1 (Public Review): https://doi.org/10.7554/eLife.84628.3.sa1
Reviewer #2 (Public Review): https://doi.org/10.7554/eLife.84628.3.sa2
Reviewer #3 (Public Review): https://doi.org/10.7554/eLife.84628.3.sa3
Author Response https://doi.org/10.7554/eLife.84628.3.sa4

## Additional files

### Supplementary files

• Supplementary file 1. Summary of experimental design, sample information, and donor demographic details. Information includes the types of assays performed, donor demographic details, sample IDs, number of Visium tissue areas per sample, and sequencing summary statistics for each sample. The table is provided as a .xlsx file.

• Supplementary file 2. Results for differential expression (DE) testing and spatially variable genes (SVGs) analyses. (A) Differential expression (DE) testing results in pseudobulked Visium SRT data. Columns include gene ID, gene name, mean log-transformed normalized counts (logcounts) in manually annotated LC and non-LC regions ('mean_logcounts_LC' and 'mean_logcounts_nonLC'), $\log_2$ fold change ($\log_2$FC), p-value, false discovery rate (FDR), and columns identifying significant (FDR <0.05 and FC >2) and highly significant (FDR <$10^{-3}$ and FC >3) genes. The table is provided as a .xlsx file sheet. (B) Spatially variable genes (SVGs) in Visium SRT data. Results for SVGs identified using nnSVG in Visium SRT data. Columns include gene ID, gene name, overall rank of SVGs identified in replicated tissue areas ('replicated_overall_rank,' i.e. top LC-associated SVGs; see Results), overall rank of identified SVGs according to average rank across tissue areas ('overall_rank,' i.e. including choroid plexus-associated SVGs from one donor; see Results), average rank of identified SVGs across individual tissue areas ('average_rank'), number of times (tissue areas) identified within top 100 SVGs ('n_withinTop100'), and ranks within each individual tissue area. The table is provided as a .xlsx file sheet. (C) DE testing results for NE neuron cluster vs. other neuronal clusters in snRNA-seq data. DE testing results comparing NE neuron cluster against other neuronal clusters in snRNA-seq data. Columns include gene ID, gene name, sum of UMI counts across all nuclei ('sum_gene'), average logcounts within the NE neuron cluster ('self_average'), average of average logcounts within other neuronal clusters ('other_average'), combined p-value, FDR, summary $\log_2$ fold change in the pairwise comparison with the lowest p-value ('summary_logFC'), and column identifying significant (FDR <0.05 and FC >2) genes. The table is provided as a .xlsx file sheet. (D) DE testing results for NE neuron cluster vs. all other clusters in snRNA-seq data. DE testing results comparing NE neuron cluster against all other clusters in snRNA-seq data. Columns include gene ID, gene name, sum of UMI counts across all nuclei ('sum_gene'), average logcounts within the NE neuron cluster ('self_average'), average of average logcounts within all other clusters ('other_average'), combined p-value, FDR, summary $\log_2$ fold change in the pairwise comparison with the lowest p-value ('summary_logFC'), and column identifying significant (FDR <0.05 and FC >2) genes. The table is provided as a .xlsx file sheet. (E) DE testing results for overlapping set of DE genes in Visium SRT data and snRNA-seq data. DE testing results for overlapping set of statistically significant (FDR <0.05 and FC >2) DE genes in pseudobulked Visium SRT data and between NE neuron cluster vs. all other clusters in snRNA-seq data. Genes are ordered by FC in Visium SRT data. Columns as in A and D. The table is provided as a .xlsx file sheet. (F) DE testing results for 5-HT neuron cluster in snRNA-seq data. DE testing results comparing 5-HT neuron clusters against all other neuronal clusters in snRNA-seq data. Columns include gene ID, gene name, sum of UMI counts across all nuclei ('sum_gene'), average logcounts within the 5-HT neuron cluster ('self_average'), average of average logcounts within all other neuronal clusters ('other_average'), combined p-value, FDR, summary $\log_2$ fold change in the pairwise comparison with the lowest p-value ('summary_logFC'), and column identifying significant (FDR <0.05 and FC >2) genes. The table is provided as a .xlsx file sheet.

• Supplementary file 3. Combined PDF file of figure supplements. Combined PDF file containing copies of all figure supplements.

• MDAR checklist

### Data availability

The datasets described in this manuscript are freely accessible in web-based formats from https://libd. shinyapps.io/locus-c_Visium/ (Shiny app containing Visium SRT data; *Chang et al., 2019*) and https:// libd.shinyapps.io/locus-c_snRNA-seq/ (iSEE app containing snRNA-seq data; *Rue-Albrecht et al., 2018*), and in R/Bioconductor formats from https://bioconductor.org/packages/WeberDivechaLC-data (R/Bioconductor ExperimentHub data package containing Visium SRT data in SpatialExperiment (*Righelli et al., 2022*) format and snRNA-seq data in SingleCellExperiment format [*Amezquita et al., 2020*])(*Weber and Divecha, 2023b*). All data accessible through the web apps is also included in the R/Bioconductor data package (*Weber and Divecha, 2023b*). The R/Bioconductor data package is available in Bioconductor version 3.16 onwards. Instructions to install and access the R/Bioconductor data package are also available from GitHub at https://github.com/lmweber/WeberDivechaLCdata (*Weber and Divecha, 2023b*). Raw data files (FASTQ sequencing data files and high-resolution TIF image files) are available via Globus from the WeberDivecha2023_locus_coeruleus data collection from the jhpce#globus01 Globus endpoint, which is also listed at http://research.libd.org/globus/. The Globus repository is not publicly accessible due to individually identifiable donor genetic variants in the FASTQ files. Users may request access to the Globus repository by submitting an email request to the corresponding authors outlining the proposed use of the data. Access will be granted for research purposes, including academic and commercial. All code used to analyze the data in the Globus repository is included in the code repository described in Code Availability. Deidentified data that may be accessed without restrictions is included in the R/Bioconductor data package described above. A combined PDF file containing copies of all supplementary figures is provided in *Supplementary file 3*.

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
