## [Editor Report · eLife assessment]

This is an **important** initial study of cell type and spatially resolved gene expression in and around the locus coeruleus, the primary source of the neuromodulator norepinephrine in the human brain. The data are generated with cutting-edge techniques, and the work lays the foundation for future descriptive and experimental approaches to understand the contribution of the locus coeruleus to healthy brain function and disease. The empirical support for the main conclusions is **solid**. This paper, and the associated web application, will be of great interest to neuroscientists working on arousal-based behaviors and neurological and neuropsychiatric phenotypes.

---

## [Referee Report · Reviewer #1 (Public Review)]

Weber et al. collect locus coeruleus (LC) tissue blocks from 5 neurotypical European men, dissect the dorsal pons around the LC, and prepare 2-3 tissue sections from each donor on a slide for 10X spatial transcriptomics. From three of these donors, they also prepared an additional section for 10x single nucleus sequencing. Overall, the results validate well-known marker genes for the LC (e.g. DBH, TH, SLC6A2), and generate a useful resource that lists genes that are enriched in LC neurons in humans, with either of these two techniques. A comparison with publicly available mouse and rat datasets identifies genes that show reliable LC enrichment across species. Their analyses also support recent rodent studies that have identified subgroups of interneurons in the region surrounding the LC, which show enrichment for different neuropeptides. In addition, the authors claim that some LC neurons co-express cholinergic markers and that a population of serotonin (5-HT) neurons is located within or near the LC. These last two claims must be taken with great caution, as several technological limitations restrict the interpretation of these results. Technical limitations currently limit the ability to integrate spatial and single-nucleus sequencing, yet the manuscript presents a valuable resource on the gene expression landscape of the human LC.

---

## [Referee Report · Reviewer #2 (Public Review)]

The data generated for this paper provides an important resource for the neuroscience community. The locus coeruleus (LC) is the known seed of noradrenergic cells in the brain. Due to its location and size, it remains scarcely profiled in humans. Despite the physically minute structure containing these cells, its impact is wide-reaching due to the known neuromodulatory function of norepinephrine (NE) in processes like attention and mood. As such, profiling NE cells has important implications for most neurological and neuropsychiatric disorders. This paper generates transcriptomic profiles that are not only cell-specific but which also maintain their spatial context, providing the field with a map for the cells within the region.

Strengths:

Using spatial transcriptomics in a morphologically distinct region is a very attractive way to generate a map. Overlaying macroscopic information, i.e. a region with greater pigmentation, with its corresponding molecular profile in an unbiased manner is an extremely powerful way to understand the specific cellular and molecular composition of that brain structure.

The technologies were used with an astute awareness of their limitations, as such, multiple technologies were leveraged to paint a more complete and resolved picture of the cellular composition of the region. For example, the lack of resolution in the spatial transcriptomic platform was compensated by complementary snRNA-seq and single molecule FISH.

This work has been made publicly available and accessible through a user-friendly application such that any interested researcher can investigate the level of expression of their gene of interest within this region.

Two important implications from this work are (1) the potential that the gene regulatory profiles of these cells are only partially conserved across species, humans, and rodents, and (2) that there may be other neuromodulatory cell types within the region that were otherwise not previously localized to the LC

Weaknesses:

Given that the markers used to identify cells are not as specific as they need to be to definitively qualify the desired cell type, the results may be over-interpreted. Specifically, TH is the primary marker used to qualify cells as noradrenergic, however, TH catalyzes the synthesis of L-DOPA, a precursor to dopamine, which in turn is a precursor for epinephrine and norepinephrine suggesting some of the cells in the region may be dopaminergic and not NE cells. Indeed, there are publications to support the presence of dopaminergic cells in the LC (see Kempadoo et al. 2016, Takeuchi et al., 2016, Devoto et al. 2005). This discrepancy is further highlighted by the apparent lack of overlap per given Visium spots with TH, SCL6A2, or DBH. While the single-nucleus FISH confirms that some of the cells in the region are noradrenergic, others very possibly represent a different catecholamine. As such it is suggested that the nomenclature for the cells be reconsidered.

The authors are unable to successfully implement unsupervised clustering with the spatial data, this greatly reduces the impact of the spatial technology as it implies that the transcriptomic data generated in the study did not have enough resolution to identify individual cell types.

The sample contribution to the results is highly unbalanced, which consequently, may result in ungeneralizable findings in terms of regional cellular composition, limiting the usefulness of the publicly available data.

This study aimed to deeply profile the LC in humans and provide a resource to the community. The combination of data types (snRNA-seq, SRT, smFISH) does in fact represent this resource for the community. However, due to the limitations, of which, some were described in the manuscript, we should be cautious in the use of the data for secondary analysis. For example, some of the cellular annotations may lack precision, the cellular composition also may not reflect the general population, and the presence of unexpected cell types may represent the accidental inclusion of adjacent regions, in this case, serotonergic cells from the Raphe nucleus.

Nonetheless having a well-developed app to query and visualize these data will be an enormous asset to the community especially given the lack of information regarding the region in general.

---

## [Referee Report · Reviewer #3 (Public Review)]

In this study, the authors present the first comprehensive transcriptome map of the human locus coeruleus using two independent but complementary approaches, spatial transcriptomics and single-nucleus RNA sequencing. Several canonical features of locus coeruleus neurons that have been described in rodents were conserved, but potentially important species differences were also identified. This work lays the foundation for future descriptive and experimental approaches to understanding the contribution of the locus coeruleus to healthy brain function and disease.

This study has many strengths. It is the first reported comprehensive map of the human LC transcriptome and uses two independent but complementary approaches (spatial transcriptomics and snRNA-seq). Some of the key findings confirmed what has been described in the rodent LC, as well as some intriguing potential genes and modules identified that may be unique to humans and have the potential to explain LC-related disease states. The main limitations of the study were acknowledged by the authors and include the spatial resolution probably not being at the single cell level and the relatively small number of samples (and questionable quality) for the snRNA-seq data. Overall, the strengths greatly outweigh the limitations. This dataset will be a valuable resource for the neuroscience community, both in terms of methodology development and results that will no doubt enable important comparisons and follow-up studies.

---

## [Author Response]

The following is the authors’ response to the original reviews.

**Reviewer #1 (Recommendations for the Authors):**
The authors provide their data and code via Github, and that shiny apps allow easy access to their data. However, spending a few minutes with the snRNAseq app I could not figure out how to search for individual genes (e.g. DBH) on their web interface. Some changes could help to make this app more user-friendly.

While it was not possible to easily modify the user interface of the snRNA-seq app itself, we have instead added two additional supplementary figures displaying screenshots and schematics with sequential instructions that provide a short tutorial showing how to search for individual genes and display either spatial gene expression (for the Visium SRT data) or gene expression by cluster or population (for the snRNA-seq data) in each interactive web app (Figure 3-figure supplement 20-21). We hope this makes the apps more accessible and assists users to more easily query specific genes that they are interested in.

The first sentence of the abstract and line 70 on page 2 need to be revised for language / grammar / clarity.

We have revised these two sentences. Line 70 on page 2 contained a typo / copy-paste error. Thank you for pointing this out.

**Reviewer #2 (Recommendations For The Authors):**
While the efforts of the authors to identify NE neurons in the LC is appreciated, the data fall a little short of conclusively calling these neurons solely noradrenergic as there is an apparent lack of overlap between TH and SLC6A2 in the spots. Undoubtedly, some spots contain both which is consistent with the RNA scope results, but there is clearly a pattern that shows spots that don't contain both. It would be worth testing the presence of other catecholamines in some of these certain spots particularly dopamine (Kempadoo et al. 2016, Takeuchi et al., 2016, Devoto et al. 2005).

We agree this is an important point. To more rigorously investigate whether TH is co-expressed within cells that produce other catecholamines, particularly dopamine (DA) in addition to norepinephrine (NE), we have included additional analyses of the snRNA-seq and Visium data, as well as generated additional RNAscope data in the revised manuscript, as follows.

(i) We investigated the spatial expression of DA neuron marker genes besides TH, including SLC6A3 (encoding the dopamine transporter), ALDH1A1, and SLC26A7 in the Visium samples (Figure 3-figure supplement 15), which shows that these genes are not strongly expressed within the manually annotated LC regions in the Visium samples (see Figure 2-figure supplement 1).

(ii) We investigated expression of DA neuron marker genes SLC6A3, ALDH1A1, and SLC26A7 in the snRNA-seq clustering (updated heatmap in Figure 3-figure supplement 8), which shows minimal expression of these genes within the NE neuron cluster (cluster 6).

(iii) Despite the data above suggesting little expression of markers for DA neurons within the human LC, we wanted to investigate this question more thoroughly with an orthogonal method given that relatively lower coverage in the sequencing approaches may miss expression, particularly for more lowly expressed transcripts. We generated new high-resolution RNAscope smFISH images at 40x magnification for samples from 3 additional donors (Br8689, Br5529, and Br5426) showing expression of NE neuron marker genes (DBH and TH), a 5-HT neuron marker gene (TPH2), and a DA neuron marker gene (SLC6A3) within individual cells within the LC regions in these samples. Expression of SLC6A3 within individual NE neurons (identified by co-expression of DBH and TH) was not apparent in these RNAscope images (Figure 3-figure supplement 16).

Together with the previous high-magnification RNAscope images showing co-expression of NE neuron marker genes (DBH, TH, and SLC6A2) within individual NE neurons (Figure 3-figure supplement 4), these new results further strengthen the conclusion that the observed TH+ cells we profiled in the LC are NE-producing neurons. In our view, the lack of observed co-expression of TH and SLC6A2 within some individual Visium spots is likely due to sampling variability and relatively lower sequencing coverage in the Visium data, rather than a true lack of co-expression. We have included additional text in the Results and Discussion further discussing this issue.

Likewise, given the low throughput of RNA scope, and the fact that it was not done in a systematic manner, it does not conclusively identify the cell types in the region. It might be worth a systematic survey of the cells in the region with both NE and DA markers. Otherwise, it is suggested that the authors be more conservative with their annotations.

As discussed above, we have now generated additional high-magnification RNAscope images for 3 independent donors (Br8689, Br5529, and Br5426), visualizing expression of two NE neuron marker genes (DBH and TH), one 5-HT neuron marker gene (TPH2), and one DA neuron marker gene (SLC6A3, encoding the dopamine transporter) within individual cells within the LC region in each sample (Figure 3-figure supplement 16). Expression of the DA neuron marker gene (SLC6A3) within individual NE neuron cell bodies (identified by co-expression of DBH and TH) was not apparent in these RNAscope images. Together with our previous RNAscope images showing co-expression of DBH, TH, and SLC6A2 within individual cells (Figure 3-figure supplement 4), in our view, these results provide strong evidence that the observed TH+ cells in the LC are NE-producing neurons, and the data do not provide supporting evidence for the existence of DA-synthesizing neurons in the human LC.

For the manual annotation, it would be useful to include HE tissue images to better understand how the annotations were derived especially because the annotations are not well corroborated by the clustering.

We have now included the H&E stained histology images for the Visium samples in Figure 2-figure supplement 2A, which can be compared with the previous figures showing the manual annotations for the LC regions (Figure 2-figure supplement 1). The histology images can also be viewed at higher resolution through the Shiny web app (https://libd.shinyapps.io/locus-c_Visium/).

The unsupervised clustering is certainly contingent on the number of genes detected, which is in turn dependent on the quality of the material and the success of the experiment. It is unclear from the methods whether the samples were pooled for clustering. If they were pooled, the author might consider using only the samples with UMIs > 500. The low UMI may represent free-floating RNA, suggesting issues with tissue permeabilization in turn influencing the ability to confidently associate genes with spots. Sticking with the higher quality sample may improve the ability to perform unsupervised clustering.

For the spot-level unsupervised clustering using BayesSpace, our aim was to demonstrate whether it is feasible to segment the LC and non-LC regions in the Visium samples in a data-driven manner using a spatial clustering algorithm, instead of relying on manual annotations. We performed clustering across samples (i.e. pooled) -- we have included additional wording in the text and figure caption to clarify this. We agree with the reviewer there may be further optimizations possible, such as filtering out spots or samples with low UMI counts. However, filtering out low-UMI spots may also confound the clustering if low-UMI spots are associated with biological signal (e.g. preferentially located in white matter regions). Overall, we found that applying data-driven methods such as BayesSpace to segment the LC and non-LC regions did not perform sufficiently to rely on for our downstream analyses (Figure 2-figure supplement 6), and, in our view, further incremental optimizations were unlikely to reach sufficient performance and robustness, so we chose to rely on the manual annotations instead. In addition, as noted in the Results, this avoids potentially inflated false discoveries due to issues of circularity whenperforming differential gene expression testing between regions defined by unsupervised clustering on the same sets of genes (Gao et al. 2022). We included the BayesSpace results (Figure 2-figure supplement 6) to provide information and ideas to method developers interested in using this dataset as a test case for further development of spatial clustering algorithms. However, further adapting or optimizing these spatial clustering algorithms ourselves was not within the scope of our current work.

It is not entirely clear why the authors used FANS, especially with the scored tissue. Do the authors think this could have negatively influenced the capture of the desired cell type since FANS can compromise the integrity of the nuclei? In other words, have the authors considered that this may have resulted in a loss rather than enrichment? The proportion of "NE" neurons in the snRNA-Seq data is less than 2% in all cases and at its lowest in sample 6522 which does not correspond well with the proportion of tissue that was manually annotated as containing NE cells, even when taken into consideration the potential size difference of cells. In the same vein, in some samples, there are more "5-HT" neurons in the region than "NE" according to the numbers.

As noted in our initial response to reviewers (“Response to Public Review Comments”), we used FANS to enrich for neurons based on our previous success with this approach to identify relatively rare neuronal populations in other brain regions (e.g. nucleus accumbens and amygdala; Tran and Maynard et al. 2021). Based on this previous work, our rationale was that without neuronal enrichment, we could potentially miss the LC-NE population, given the relative scarcity and low absolute number of this neuronal population (e.g. estimates of ~50K total in the entire human LC).

We do not have a definitive answer to the question of whether our use of FANS to enrich for neurons may have led to damage and contributed to the low recovery rate of LC-NE neurons (as well as the relatively increased levels of mitochondrial contamination compared to other brain regions / preparations in the human brain in our hands). Due to our limited tissue resources for this study, we did not have sufficient tissue to perform a direct comparison with non-sorted data. However, we agree with the reviewer that this is plausible, and warrants further investigation in future work. In particular, the relatively large size and fragility of LC-NE neurons, as well as our use of a standard cell straining approach (70 µm, which may not be ideal for this population), may also be contributing factors. Systematically optimizing the preparation to attempt to increase recovery rate (and decrease mitochondrial contamination) are important avenues for future work, and we have decided to share our data and experiences now to assist other groups performing related work. We have included additional wording in the Discussion to further highlight these issues.

The majority of the snRNA-seq remained unannotated "ambiguous" neurons. It would be highly advantageous to include an annotation for these numerous cells.

These nuclei were unidentifiable due to ambiguous marker gene expression profiles, i.e. expression of pan-neuronal marker genes without clear expression of either excitatory or inhibitory neuronal marker genes (see Figure 3A and Figure 3-figure supplement 8). Since we were not able to clearly identify these clusters, and due to our additional concerns regarding the data quality (e.g. low recovery rate of the NE neuron population of interest, potential cell damage, and mitochondrial contamination), we decided to label these neuronal clusters as “ambiguous” instead of assigning low-confidence cluster labels. We have included additional wording in the Results section to explain this issue.

The most likely explanation for identifying serotonergic neurons in these samples is the inclusion of the Raphe Nucleus within the dissection, especially since these cells do not map to the LC per se. As such, is there a way to neuroanatomically define the potential inclusion of this region from these tissue blocks used? Or to the contrary, definitively demonstrate the exclusion of the Raphe?

As noted in our initial response to reviewers (“Response to Public Review Comments”), our dissection strategy in this initial study precluded the ability to keep track of the exact orientation of the tissue sections on the Visium arrays with respect to their location within the brainstem. Therefore, it is not possible to definitively answer the question of whether the dissections included the raphe nucleus, and if so, which portion of it, based on neuroanatomy from the tissue blocks.

However, during the course of this study and in parallel, ongoing work for other small, challenging brain regions, we developed a number of specialized technical and logistical strategies for keeping track of orientation and mounting serial sections from the same tissue block onto a single spatial array, which is extremely technically challenging. We are now well-prepared for addressing these issues in future studies, e.g. keeping track of the orientation of the dissections and potential inclusion of adjacent neuroanatomical structures. We have included additional details on this issue in the Discussion.

Given that one sample (Visium capture area) was excluded as it did not seem to contain a representation of the LC for the profiling of "NE" cells, does it make sense to include this sample in the analysis of 5HT cells given the authors are trying to make claims about the cell composition in and around the LC? Since there appears to be little 5HT contribution from this sample and its inclusion results in inconsistency across experiments and not any notable advantages, the authors might want to reconsider its inclusion in the results.

We identified a cluster of 5-HT neurons in the snRNA-seq data (Figure 3) and used the Visium samples to further investigate the spatial distribution of this population (Figure 3-figure supplement 9). For the enrichment analyses in the Visium data (Figure 3-figure supplement 9C), we used only the 8 Visium samples that passed quality control (QC). We included the 9th sample (which did not pass QC) in the spot plot visualizations (Figure 3-figure supplement 9A-B) for completeness, but did not base our main conclusions on this sample (in this sample, the tissue resource was likely depleted during earlier sections, so the section for the Visium sample was taken slightly past the extent of the LC within this tissue block). We have included additional wording in the Results section and figure captions to clarify this issue.

For the RNAscope images, it would be useful to include (draw) the manual annotation of the LC to facilitate interpretation. This is especially useful for demonstrating the separate populations of 5HT and "NE" cells. In general, it would be useful to keep a hashed line perimeter for all sections processed by Visium.

We have now added a dashed outline indicating the manually annotated LC region in the RNAscope image showing the full tissue section (Figure 3-figure supplement 11). The high-magnification RNAscope images (Figure 3-figure supplement 4, 16, and 17) show regions entirely within the LC regions -- we have included additional wording to note this in the figure captions. For the Visium spot plots, we either labeled spots within the annotated regions within the figures or included additional wording in the figure captions to refer to the figures showing the annotations (Figure 2-figure supplement 1).

The authors state that they successfully mapped the NE neuron population from snRNA-seq to the manually annotated regions on the Visium slides. Based on the color-coded map, these results are not very convincing since the abundance of the given transcript profile is extremely low. Here again, it would help to draw a hashed line perimeter on the slide to denote the manually annotated region.Perhaps the authors could try a different strategy for mapping snRNA signal to the slide? However, it appears that the mapping worked better for the capture areas with higher UMI/genes counts. Perhaps the authors should consider using only the slides with high gene/UMI counts.

We agree that the performance of these analyses (Figure 3-figure supplement 14) was not clearly described in the previous version of the manuscript. We have rewritten the corresponding paragraph in the Results section to make it more clear that the mapping (spot-level deconvolution) performance was relatively poor overall, and that we did not use these results for further downstream analyses. We did however want to include these results from the cell2location algorithm to provide information and data for method developers on the challenges of these types of analyses in our dataset (e.g. due to the presence of rare populations, relatively subtle differences in expression profiles between neuronal subpopulations, and potential issues due to large nuclei size and high transcriptional activity for NE neurons). While further approaches for these types of analyses exist, and additional optimizations such as subsetting samples or spots with high UMI counts could also be investigated, in our view, these further optimizations lie outside the scope of our current work. We have also added wording in the figure caption to refer to Figure 2-figure supplement 1, which displays the corresponding annotated LC regions per sample.

It is hard to see if the RNA scope image Supplementary Figure 11 shows co-localization of SLC6A2, TH, and DBH. Having the individual image from each microscope filter along with the merged image is required to properly assess the colocalization of the signals.

We updated the multi-channel RNAscope images to show both the merged channels and individual channels in separate panels (Figure 3-figure supplement 4, 16, and 17), which makes the visualization more clear. Thank you for this suggestion. (Note that the previous Supplementary Figure 11 has been re-numbered to Figure 3-figure supplement 4.)

The heatmap showing the level of marker transcripts shows a much lower expression of specific markers, TH, DBH, SLC6A2 in NE vs other clusters looks surprisingly low (particularly TH), while the much broader marker SLC18A2 (monoamine transporter) is considerably more differential. What do the authors make of this finding?

This is correct. In the snRNA-seq data, we observed that SLC18A2 is one of the most highly differentially expressed (DE) genes in the NE neuron cluster vs. other neuronal clusters, with a high level of expression in the NE neuron cluster (Figure 3C). Note that this heatmap shows the top 70 DE genes (excluding mitochondrial genes) out of the full list of 327 statistically significant DE genes with elevated expression in the NE neuron cluster (the full list of 327 genes is provided in Supplementary File 2C). While all four of these genes (DBH, TH, SLC6A2, and SLC18A2) are identified as statistically significant DE genes, SLC18A2 is the most highly DE out of these and has an especially high level of expression in the NE neuron cluster, as noted by the reviewer (Figure 3C). This could be due to the fact that SLC18A2 transcripts are expressed at higher absolute levels in these neurons than the transcripts that are more specific to LC-NE neurons. While it is true that SLC18A2 is a “broader” marker in the sense that it is found in more cell types -- e.g. cell types within brain nuclei that contain monoaminergic as well as brain nuclei that contain catecholaminergic cells -- expression of SLC18A2 within the LC is highly specific to the catecholaminergic LC-NE neurons given its specialized functional role within monoamine and catecholamine neurons in packaging amine neurotransmitters into synaptic vesicles. We note that SLC18A2 plays a specialized role that is critical to the core function of LC-NE neurons, and hence we are not particularly surprised with this finding and think that one possibility is that this differential expression appears more robustly due to higher absolute levels of the marker.

While it is understandable that the authors decided to include cells/nuclei with high mitochondrial reads, further work is needed to ensure these cells are of sufficient quality to use in an unbiased way knowing that a high percentage of mitochondrial reads in nuclei sequencing is usually indicative of low-quality nuclei. This can be assessed by evaluating the quality of the nuclei with GWA, which stains an intact nuclear membrane acting as a measure of the integrity of the nuclei.

To further investigate these results, we added additional analyses evaluating quality control (QC) metrics for the NE neuron cluster in the snRNA-seq data, which had an unusually high proportion of mitochondrial reads (Figure 3-figure supplement 2, shown also below in comments for Reviewer 3) (see also related Figure 3-figure supplement 1, 3, which were included in the manuscript previously). These additional QC analyses do not show any other problematic values for this cluster, other than the high mitochondrial proportion, so we do not believe this is purely a data quality issue. We are aware that this is an unexpected result -- in most cell populations, a high proportion of mitochondrial reads would be indicative of cell damage and poor data quality. However, we have recently also observed high mitochondrial proportions in other relatively rare neuronal populations characterized by large size and high metabolic demand. As discussed below for Reviewer 3, we believe that this is mitochondrial “contamination”, as there should be no mitochondrial reads per se within the nuclear compartment. However, it may be possible that in cell populations that have abundant levels of mitochondria and high transcript expression of mitochondrial transcripts in the cell body, that the likelihood of ambient RNA capture of mitochondrial transcripts during nuclear preparation may be higher than for other cell types that have lower expression of mitochondrial transcripts. Hence, we believe that our interpretation is likely correct, i.e. that a combination of technical and biological factors contributes to the inclusion of a relatively high amount of mitochondrial RNA within the droplets for these nuclei. We agree with the reviewer that this finding warrants further investigation in future work. However, in our current study, the tissue resource is depleted for any further experimental validation of this question, so we preferred to provide our data to the community in its current form, while transparently noting this unexpected finding in our results. We have included additional text in the Results section describing the new QC analyses shown in Figure 3-figure supplement 2.

Minor comments:Line 319-321 could be written more clearly to indicate that due to the lack of resolution in a given spot, there are "contaminating reads" that reduce the precision of the cell profile. This reduced precision is likely what results in the "lack of conservation" across species.

We have added additional wording to this sentence to clarify this point.

In the discussion, the authors write that the analyses "unbiasedly identified a number of genes enriched in human LC", however, given the manual annotation of the region for each capture area, this resulted in a biased assessment of the spots.

We have replaced this wording to refer to “untargeted, transcriptome-wide” analyses (i.e. analyses that are not based on a targeted panel of genes) instead of “unbiased”. We agree that the meaning of “unbiased” is ambiguous in this context.

**Reviewer #3 (Recommendations For The Authors):**
Major points:Overall, the discovery of some cells in the LC region that express serotonergic markers is intriguing. However, no evidence is presented that these neurons actually produce 5-HT. Perhaps more conservative language would be appropriate (i.e. "cells that possess mRNA signatures of serotonergic neurons" or something like that). Did these cells co-express other markers one would expect in 5-HT neurons like 5-HT autoreceptors and SLC6A18? Also would be useful to compare expression profiles of these putative 5-HT neurons with any published material on bona fide dorsal raphe 5-HT neurons. For the RNAscope confirmation in the supplementary material, it would be helpful to show each marker separately as well as the overlay, and to include representative higher magnification images like were provided for the ACH markers.

Thank you for this comment. In order to further investigate the identity of these cells, we have investigated the expression of several additional genes including SLC6A18, 5-HT autoreceptor genes (HTR1A, HTR1B), marker genes for 5-HT neurons (SLC18A2, FEV), and marker genes for 5-HT neuronal subpopulations within the dorsal and median raphe nuclei from the literature (Ren et al. 2019), in both the Visium and the snRNA-seq data.

We observed some expression of SLC18A2 and FEV within the same areas as SLC6A4 and TPH2 in the Visium samples (Figure 3-figure supplement 10A-B, reproduced below; note that SLC18A2 is also a marker gene for NE neurons located within the LC regions), consistent with Ren et al. (2019). However, we did not observe a strong or consistent expression signal for the 5-HT autoreceptors (HTR1A, HTR1B) (Figure 3-figure supplement 10C-D, reproduced below), and we observed zero expression of SLC6A18 in the Visium samples. In the snRNA-seq data, within the cluster identified as 5-HT neurons, we observed some expression of SLC18A2, low expression of FEV, and almost zero expression of SLC6A18 (Figure 3-figure supplement 8, reproduced below; note that SLC6A18 is not shown since it was removed during filtering for low-expressed genes). Similarly, we observed very low expression of the 5-HT autoreceptors (HTR1A, HTR1B) and the additional marker genes for 5-HT neuronal subpopulations from Ren et al. (2019) -- with the possible exception of the neuropeptide receptor gene HCRTR2, which was identified by Ren et al. (2019) within several clusters in both the dorsal and median raphe in mice (Figure 3-figure supplement 8, reproduced below).

Overall, these additional results give us some further confidence that these are likely 5-HT neurons (due to expression of SLC18A2 and FEV), while also raising further questions (due to the absence of 5-HT autoreceptor genes HTR1A, HTR1B and 5-HT neuronal subpopulation marker genes). While we believe that the most likely explanation is the inclusion of 5-HT neurons from the edges of the adjacent dorsal raphe nuclei in our samples, we acknowledge that the evidence presented is not fully conclusive and does not identify specific subpopulations of 5-HT neurons. In addition, the limited size of our dataset (number of samples and cells) and the lack of information on sample orientation precludes any definitive identification of subpopulations based on their association with specific anatomical regions within the dorsal raphe nuclei. We have updated the manuscript by (i) adjusting our language in the Results and Discussion, (ii) including the additional analyses, supplementary figures, and reference to the literature (Ren et al. 2019) discussed above, and (iii) including additional wording in the Discussion on improvements to the dissection strategy that would allow these questions to be addressed in future studies via a focused molecular profiling of the dorsal raphe nuclei across the rostral-caudal axis.

Regarding the RNAscope images, we have included additional images showing channels side-by-side and higher magnification, as suggested (and also discussed above for Reviewers 1 and 2). In addition, we have added an outline highlighting the LC region in Figure 3-figure supplement 11 (as suggested above by Reviewer 2), and included an additional high-magnification RNAscope image demonstrating co-expression of 5-HT neuron marker genes (TPH2 and SLC6A4) within individual cells (Figure 3-figure supplement 12).

Concerning the snRNA-seq experiments, why were only 3 of the 5 donors used, particularly given the low number of LC-NE nuclear transcriptomes obtained? How were the 3 donors chosen from the 5 total donors and how many 100 um sections were used from each donor? Are the 295 nuclei obtained truly representative of the LC population or are they just the most resilient LC nuclei? How many LC nuclei would be estimated to be captured from staining the 100 um tissue sections?

As discussed in our previous response to reviewers (“Response to Public Review Comments”), the reason we included only 3 of the 5 donors for the snRNA-seq assays was due to tissue availability on the tissue blocks. In this study, we were working with a finite tissue resource. Due to the logistics and thickness of the required tissue sections for Visium (10 μm) and snRNA-seq (100 μm), running Visium first allowed us to ensure that we could collect data from both assays -- if we ran snRNA-seq first and captured no neurons, the tissue block would be depleted. Due to resource depletion, we did not have sufficient available tissue remaining on all tissue blocks to run the snRNA-seq assay for all donors. We have conducted extensive piloting in other brain regions on the amount (mg) of tissue that is needed from various sized cryosections, and the LC is particularly difficult since these are small tissue blocks and the extent of the structure is small. Hence, in some of the subjects, we did not have sufficient tissue available for the snRNA-seq assay.

We have included details on the number of 100 μm sections used for each donor in Methods -- this varied between 10-15 sections per donor, approximating 50-80 mg of tissue per donor.

Regarding the question about the representativeness / resilience of the LC nuclei -- as discussed in our previous response to reviewers (“Response to Public Review Comments”) and above for Reviewer 2, we agree that this is a concern. As discussed above for Reviewer 2, it is plausible that our use of FANS may have contributed to cell damage and the low recovery rate of LC-NE neurons. The relatively large size and fragility of LC-NE neurons, as well as our use of a standard cell straining approach (70 µm, which may not be ideal for this population), may also be contributing factors. Due to our limited tissue resource, we did not have sufficient tissue to perform a direct comparison with non-sorted data. Systematically optimizing the preparation to attempt to increase recovery rate is an important avenue for future work. We have included additional discussion of this issue in the Discussion.

Regarding the question about the number of expected nuclei, we have now included estimates of the number of cells per spot within the LC regions in the Visium data (see also related point below, and Figure 2-figure supplement 2B reproduced below), based on the H&E stained histology images and use of cell segmentation software (VistoSeg; Tippani et al. 2022). While we do not have any confident estimates of the number of expected nuclei in the snRNA-seq data, these estimates of cell density from the Visium data could, together with information on additional factors such as the accuracy of the tissue scoring and the effectiveness of FANS, be used to help derive an an expected number of nuclei in future studies. We have included additional wording in the Discussion to note that these estimates could be used in this manner during future studies.

The LC displays rostral/caudal and dorsal/ventral differences, including where they project, which functions they regulate, and which parts are vulnerable in neurodegenerative disease (e.g. Loughlin et al., Neuroscience 18:291-306, 1986; Dahl et al., Nat Hum Behav 3:1203-14, 2019; Beardmore et al., J Alzheimer's Dis 83:5-22, 2021; Gilvesy et al., Acta Neuropathol 144:651-76, 2022; Madelung et al., Mov Disord 37:479-89, 2022). Which part(s) of the LC was captured for the SRT and snRNAseq experiments?

As discussed in our previous response to reviewers (“Response to Public Review Comments”), a limitation of this study was that we did not record the orientation of the anatomy of the tissue sections, precluding our ability to annotate the tissue sections with the rostral/caudal and dorsal/ventral axis labels. We agree with the reviewer that additional spatial studies, in future work, could offer needed and important information about expression profiles across the spatial axes (rostral/caudal, ventral/dorsal) of the LC. Our study provides us with insight about optimizing the dissections for spatial assays, as well as bringing to light a number of technical and logistical issues that we had not initially foreseen. For example, during the course of this study and parallel, ongoing work in other, small, challenging regions, we have now developed a number of specialized technical and logistical strategies for keeping track of orientation and mounting serial sections from the same tissue block onto a single spatial array, which is extremely technically challenging. We are now well-prepared for addressing these issues in future studies with larger numbers of donors and samples in order to make these types of insights. We have included additional details in the Discussion to further discuss this point.

The authors mention that in other human SRT studies, there are typically between 1-10 cells per expression spot. I imagine that this depends heavily on the part of the brain being studied and neuronal density. In this specific case, can the authors estimate how many LC cells were contained in each expression spot?

We have now performed additional analyses to provide an estimate of the number of cells per spot in the Visium data (Figure 2-figure supplement 2B), based on the application of cell segmentation software (VistoSeg; Tippani et al. 2022) to identify cell bodies in the H&E stained histology images. We applied this methodology and calculated summary statistics within the annotated LC regions for 6 samples (see Methods), and found that the median number of cells per spot within the LC regions ranged from 2 to 5 per sample. We note that these estimates include both NE neurons and other cell types within the LC regions, and that applying cell segmentation software in this brain region is particularly challenging due to the wide range in cell body sizes, with NE neurons being especially large. We have included these updated estimates in the Results and Discussion, and additional details in Methods.

Regarding comparison of human LC-associated genes with rat or mouse LC-associated genes (Fig. 2D-F), the authors speculate that the modest degree of overlap may be due to species differences between rodent and human and/or methodological differences (SRT vs microarray vs TRAP). Was there greater overlap between mouse and rat than between mouse/rat and human? If so, that is evidence for the former. If not, that is evidence for the latter. Also would be useful for more in-depth comparison with snRNA-seq data from mouse LC. https://www.biorxiv.org/content/10.1101/2022.06.30.498327v1

Our comparisons with the mouse (Mulvey et al. 2018) and rat (Grimm et al. 2004) data showed that we observed a relatively higher overlap between the human vs. mouse data than the human vs. rat data (Figures 2F-G and 3D-E). However, we note that the substantially different technologies used(TRAP-seq in mouse vs. laser capture microdissection and microarrays in rat) make it difficult to confidently interpret the degree of overlap between the two studies, and a direct comparison of these alternative platforms (TRAP-seq vs. LCM / microarray) or species (mouse vs. rat) lies outside the scope of our study. We have included updated wording in the Results and Discussion to explain this issue and help interpret these results.

Regarding the newer mouse study using snRNA-seq (Luskin and Li et al. 2022), we have extended our analyses to perform a more in-depth comparison with this study. Specifically, we have evaluated the expression of an additional set of GABAergic neuron marker genes from this study within our secondary clustering of inhibitory neurons in the snRNA-seq data (Figure 3-figure supplement 13B). We observe some evidence of cluster-specific expression of several genes, including CCK, PCSK1, PCSK2, PCSK1N, PENK, PNOC, SST, and TAC1. We have also included additional text describing these results in the Results section.

The finding of ACHE expression in LC neurons is intriguing. Susan Greenfield has published a series of papers suggesting that ACHE has functions independent of ACH metabolism that contributes to cellular vulnerability in neurodegenerative disease. This might be worth mentioning.

We thank the reviewer for pointing this out. We were very surprised too by the observed expression of SLC5A7 and ACHE in the LC regions (Visium data) and within the LC-NE neuron cluster (snRNA-seq data), coupled with absence of other typical cholinergic marker genes (e.g. CHAT, SLC18A3), and we do not have a compelling explanation or theory for this. Hence, the work of Susan Greenfield and colleagues suggesting non-cholinergic actions of ACHE, particularly in other catecholaminergic neuron populations (e.g. dopaminergic neurons in the substantia nigra) is very interesting. We have included references to this work and how it could inform interpretation of this expression (Greenfield 1991; Halliday and Greenfield 2012) in the Discussion.

High mitochondrial reads from snRNA-seq can indicate lower quality. Can the authors comment on this and explain why they are confident in the snRNA-seq data from presumptive LC-NE neurons?

As mentioned above for Reviewer 2, we have included additional analyses to further compare quality control (QC) metrics for the NE neuron cluster (which had an unusually high proportion of mitochondrial reads) against other neuronal and non-neuronal clusters and nuclei in the snRNA-seq data (Figure 3-figure supplement 2). These additional QC analyses do not show any other problematic values for this cluster. Specifically, we show that the QC metric values for sum UMIs and detected genes per droplet for the NE neuron cluster fall within the range for (A) other neurons and (B) all other nuclei (excluding droplets with ambiguous / unidentifiable neuronal signatures). In addition, we observe that the droplets with the highest mitochondrial percentages (>75%) (C-D), which also have unusually low number of detected genes (D), tend to be from the ambiguous category (droplets with ambiguous / unidentifiable neuronal signatures), suggesting that true low-quality droplets are correctly identified and included within the ambiguous category (e.g. consisting of a mixture of debris from partial damaged nuclei) instead of as NE neurons. Since our QC analyses for the NE neuron cluster do not show any problems other than the high mitochondrial percentage, we do not believe these are simply mis-classified low-quality droplets. We also note that we have recently observed high mitochondrial proportions in other relatively rare neuronal populations characterized by large size and high metabolic demand in human data. We believe that our interpretation is correct -- i.e. that a combination of technical and biological factors has led to the inclusion of a relatively high amount of mitochondrial RNA within the droplets for these nuclei. We have included these additional QC analyses (Figure 3-figure supplement 2) and further discussion of this issue in the Results section.

The Discussion could be expanded. Because there is a lot known and/or assumed about the LC, discussing all of it is certainly beyond the scope of this manuscript. However, perhaps the authors could pick a few more for confirmation and hypothesis generation. For example, one of the most well studied and important aspects of the LC is its regulation by neuromodulatory inputs. It would be interesting for the authors to discuss the expression of receptors for CRF, cannabinoids, orexin, galanin, 5-HT, etc, particularly when compared with the available rodent TRAP and snRNA-seq data (https://www.biorxiv.org/content/10.1101/2022.06.30.498327v1) contained some surprises, such as very low expression of CRF1 in LC-NE neurons, suggesting that the powerful activation of LC cells by CRF is indirect. Does this hold up in humans?

We have expanded the Discussion to include additional discussion and references on several points, as discussed also above. Indeed these are interesting questions and these neuromodulatory systems are all of interest in the context of signaling within the LC in terms of function of the LC-NE system. We note that the manuscript serves primarily as a data resource and will be useful in many different ways depending on the different goals and interests of the readers. This is precisely why we wanted to take the time to make accessible and easy to use tools to interrogate and visualize the data. We have provided screenshots in Author response image 1-4 from the Shiny visualization app for the Visium data (https://libd.shinyapps.io/locus-c_Visium/) querying several main receptors of the neuromodulatory systems that this reviewer is particularly interested in to illustrate how the visualization apps can readily be used to query specific genes and systems of interest.

**Author response image 1. sa4fig1:** *CRHR1*.

**Author response image 2. sa4fig2:** *CNR1*.

**Author response image 3. sa4fig3:** *OXR1*.

**Author response image 4. sa4fig4:** *GALR1*.

Minor points:Line 46 add stress responses to the key functions of LC neurons

We have added this point and included additional references to support the findings.

Line 47 add that the LC was so named "blue spot" because of its signature production of neuromelanin pigment

We have added this point.

Line 49 LC's capacity to synthesize NE is not "unique" - several other brainstem/medullary nuclei also synthesize NE (e.g. A1-A7; LC is A6)

We have updated this wording.

Line 54 Although prior evidence indicated age-related LC cell loss in people without frank neurodegenerative disease, recent studies that are better powered and used unbiased stereological methods have refuted the idea that LC neurons die during normal aging (reviewed in Matchett et al., Acta Neuropathologica 141:631-50, 2021)

We have updated this part of the Introduction to focus on cell loss in the LC in neurodegenerative disease and removed the older references describing studies that suggested LC neurons die in normal aging.

Line 62 Would also be worth mentioning the role of the LC in other mood disorders where adrenergic drugs are often prescribed, such as PTSD (e.g. prazosin), opioid withdrawal (e.g. lofexidine), anxiety and depression (e.g. NE reuptake inhibitors).

We have added additional references to these disorders and their treatment with noradrenergic drugs in the Introduction.

**Additional updates from Public Review Comments:**

We have also included the following updates, in response to additional reviewer comments received during the initial round of “Public Review Comments” and which are not already described in the responses to the “Recommendations for the Authors” above.

● We included updated wording in the Results section and Figure 1C caption to more clearly describe the number of donors included in the final SRT and snRNA-seq data used for analyses after all quality control (QC) steps (4 donors for SRT data, 3 donors for snRNA-seq data).

● Figure 3-figure supplement 1D (number of nuclei per cluster in unsupervised clustering of snRNA-seq data) has been updated to show percentages of nuclei per cluster.

● We have added comparisons between the lists of differentially expressed (DE) genes identified in the Visium and snRNA-seq data. To make these sets comparable, we have added (i) snRNA-seq DE testing results between the NE neuron cluster and all other clusters (instead of other neuronal clusters only, as shown in the main results in Figure 3) (excluding ambiguous neuronal) (Figure 3-figure supplement 6 and Supplementary File 2D), and (ii) calculated overlaps and comparisons between the sets of DE genes between the Visium data (pseudobulked LC vs. non-LC regions) and the snRNA-seq data (NE neuron cluster vs. all other clusters excluding ambiguous neuronal). This comparison generated a list of 51 genes that were identified as statistically significant DE genes (FDR < 0.05 and FC > 2) in both the Visium and the snRNA-seq data (Figure 3-figure supplement 7 and Supplementary File 2E).

**Other additional updates:**

We have added an additional data repository (Globus). Raw data files (FASTQ sequencing data files and high-resolution TIF image files) are now available via Globus from the WeberDivecha2023_locus_coeruleus data collection from the jhpce#globus01 Globus endpoint, which is also listed at http://research.libd.org/globus/. The Globus repository is not publicly accessible due to individually identifiable donor genetic variants in the FASTQ files. Approved users may request access from the corresponding authors. This data repository is listed in the Data Availability section.